# Different populations of CD11b$^+$ dendritic cells drive Th2 responses in the small intestine and colon

Johannes U. Mayer[1], Mimoza Demiri[2], William W. Agace[2,3], Andrew S. MacDonald[4], Marcus Svensson-Frej[2] & Simon W. Milling[1]

T-helper 2 (Th2) cell responses defend against parasites. Although dendritic cells (DCs) are vital for the induction of T-cell responses, the DC subpopulations that induce Th2 cells in the intestine are unidentified. Here we show that intestinal Th2 responses against *Trichuris muris* worms and *Schistosoma mansoni* eggs do not develop in mice with IRF-4-deficient DCs (IRF-4$^{f/f}$ CD11c-cre). Adoptive transfer of conventional DCs, in particular CD11b-expressing DCs from the intestine, is sufficient to prime *S. mansoni*-specific Th2 responses. Surprisingly, transferred IRF-4-deficient DCs also effectively prime *S. mansoni*-specific Th2 responses. Egg antigens do not induce the expression of IRF-4-related genes. Instead, IRF-4$^{f/f}$ CD11c-cre mice have fewer CD11b$^+$ migrating DCs and fewer DCs carrying parasite antigens to the lymph nodes. Furthermore, CD11b$^+$CD103$^+$ DCs induce Th2 responses in the small intestine, whereas CD11b$^+$CD103$^-$ DCs perform this role in the colon, revealing a specific functional heterogeneity among intestinal DCs in inducing Th2 responses.

[1] Centre for Immunobiology, Institute of Infection, Immunity and Inflammation, College of Medicine, Veterinary Medicine and Life Sciences, University of Glasgow, 120 University Place, Glasgow G12 8TA, UK. [2] Immunology Section, Lund University, Sölvegatan 19, Lund BMC D14, Sweden. [3] Section for Immunology and Vaccinology, National Veterinary Institute, Technical University of Denmark, Bülowsvej 27, Frederiksberg C 1870, Denmark. [4] Faculty of Biology, Medicine and Health, Manchester Collaborative Centre for Inflammation Research, School of Biological Sciences, The University of Manchester, 46 Grafton Street, Manchester M13 9NT, UK. Correspondence and requests for materials should be addressed to S.W.M. (email: Simon.Milling@glasgow.ac.uk).

Type 2 immunity, the typical response against parasitic or allergic stimuli, can protect against parasites or exacerbate allergic conditions[1]. Approximately, a third of the world's population are infected with parasitic worms, most of which affect the gastrointestinal tract[2]. Many of these infections develop into chronic pathologies and are an enormous global health burden. There is insufficient knowledge about how worm infections, which can selectively infect either the small intestine or the colon, are controlled by the intestinal immune system. Helminth parasite infections in both the small intestine and colon induce potent T-helper 2 (Th2) responses that can control parasite burden or lead to chronic pathologies[3].

*Schistosoma mansoni* eggs and their soluble egg antigens (schistosome egg antigen, SEA) induce potent Th2 and interferon (IFN)-γ responses, both during infection with live parasites[4,5] and in experimental models in which eggs or SEA are injected into tissues[6,7]. During the natural parasite infection, a proportion of the eggs released by intravascular adult worms become lodged in the intestinal wall and the liver, where they induce strong type 2 immune responses. These eggs are central to immunopathology associated with this infection, as they induce granulomatous inflammation and tissue fibrosis, which can lead to severe organ damage[5].

Both non-professional antigen-presenting cells, such as basophils[8] and monocyte-derived dendritic cells (DCs)[9], and conventional DCs[10,11] have been shown to have functions in the induction or maintenance of Th2 responses. However, the cells that are sufficient to induce Th2 responses in the intestine have not been clearly identified.

In the small intestine and colon, four different populations of conventional DCs can be identified, categorized by their differential expression of the integrins CD11b and CD103 (refs 12–14). These populations are present at different frequencies in the small intestine and colon[15,16], and migrate via intestinal-draining lymphatics to the mesenteric lymph nodes (MLN) to initiate T-cell responses[14]. Studies have indicated that intestinal DC populations are specialized to induce different facets of the T-cell response. For example, transcription factor IFN regulatory factor (IRF)-8-dependent intestinal CD11b⁻ CD103⁺ (CD103 single-positive (SP)) DCs have a predominant function in cross-presentation to CD8⁺ T cells and induction of intestinal Th1 responses[17,18], and IRF-4-dependent CD11b⁺ CD103⁺ (double-positive (DP)) DCs seem to drive Th17 cell differentiation in intestine-draining MLNs[13,19]. Although the function of these populations in intestinal Th2 responses is unclear, studies have demonstrated that IRF-4 expression by CD11c⁺ cells is crucial for the development of Th2 responses[20,21]. In the intestine, IRF-4 is predominantly expressed by CD11b⁺ CD103⁻ (CD11b SP) DCs and DP DCs, and IRF-4 deficiency in CD11c⁺ cells results in fewer small intestinal DP DCs, as well as the absence of DP DCs and fewer CD11b SP DCs in the draining MLNs[13].

To investigate how IRF-4-expressing DCs drive intestinal Th2 responses, we use two models of human parasite infection that drive Th2 responses in the gastrointestinal tract. We address the induction of Th2 responses *in vivo* by experimental immunization with *S. mansoni* eggs and validate our findings during live infection with the intestinal parasite *Trichuris muris*. We find that CD11b-expressing DCs are specializd to drive antigen-specific Th2 responses. Furthermore, different populations of CD11b⁺ IRF-4⁺ DCs induce Th2 responses in the small intestine and colon. DP DCs from the small intestine are the only population sufficient to drive antigen-specific Th2 responses in the small intestine-draining lymph nodes and CD11b SP DCs fulfil this function in colon-draining lymph nodes. We thus demonstrate that different DC populations have distinct functions in separate regions of the intestine, which is important for understanding how intestinal immune responses are controlled, and offers the opportunity to develop more precise therapeutic targets.

## Results

**Intestinal Th2 responses require IRF-4-positive CD11c⁺ cells.** To identify the cellular mechanisms central to the induction of Th2 responses in the intestine we developed a novel method of experimental delivery of *S. mansoni* eggs directly into intestinal tissue. Eggs were injected directly into sites where they become trapped during live infection, thus providing a refined and relevant method to investigate the Th2 responses generated against trapped and penetrating eggs in the intestine (Supplementary Fig. 1a,b). The method also allowed precise temporal control of the induction of Th2 responses against *S. mansoni* eggs in the gastrointestinal tract *in vivo*, which has not been previously possible. We found that the injection of 1,000 *S. mansoni* eggs into the subserosal tissue of the small intestine was sufficient to induce antigen-specific Th2 and IFN-γ responses in the MLNs, with the key Th2 cytokines interleukin (IL)-4, IL-5 and IL-13 induced in *in vitro* total MLN cell cultures, specifically after the restimulation with SEA 5 days after *in vivo* immunization (Fig. 1a and Supplementary Fig. 1c–e). Consistent with published findings[22], we observed no antigen-specific induction of Th17 cytokines (Supplementary Fig. 1d). Intracellular flow cytometric staining after phorbol 12-myristate 13-acetate (PMA)/ionomycin stimulation confirmed that these cytokines were produced by CD4 T cells that produced IFN-γ or had differentiated into Th2 cells (Fig. 1b and Supplementary Fig. 1f,g). To determine whether intestinal egg injection could also be used as a model of colonic Th2 induction, eggs were injected either in the small intestine or colon and the small intestine-draining MLNs (sMLNs) and colon-draining MLNs (cMLNs)[16] were harvested 5 days after immunization. Analysis of restimulated individual lymph nodes revealed increased concentrations of antigen-specific cytokines, compared with analysis of pooled MLNs (Fig. 1a). These responses were only observed in the sMLNs or cMLNs draining the respective injection sites (Fig. 1c). Thus, intestinal *S. mansoni* egg injections can be used as an experimental model to further investigate the mechanisms of Th2 induction in both tissues.

Many aspects of type 2 immunity are controlled by the transcription factor IRF-4, which controls the development of Th2 cells[23], alternatively activated macrophages[24] and CD11b-expressing DCs[25]. However, little is known about how IRF-4 regulates the induction of Th2 responses in the intestine. To determine what impact the expression of IRF-4 by antigen-presenting cells had in driving intestinal Th2 responses in the small intestine and colon, we used the IRF-4^{f/f} CD11c-cre mouse model that allows targeted deletion of IRF-4 on all CD11c⁺ cells[13], including intestinal macrophages and conventional DCs. It has been reported that CD11c-cre expression in these mice does not affect DC frequencies[26,27]. Subserosal injection of *S. mansoni* eggs into the bone marrow (BM) chimeric mice, generated by lethal irradiation and reconstitution of C57BL/6.SJL mice with BM from IRF-4^{f/f} CD11c-cre-positive (cre+) or IRF-4^{f/f} cre-negative (cre−) mice (Supplementary Fig. 2a–c), resulted in dramatically impaired Th2 responses in the MLNs of cre+ chimeras, accompanied by an elevated IFN-γ response (Fig. 1d). The loss of IRF-4-dependent cells in these animals did not affect the total number of cells in the small intestine (Supplementary Fig. 2c). The impaired Th2 responses were evident in both the small intestine and colon, demonstrating a central requirement for IRF-4⁺ CD11c⁺ cells for Th2 induction in both organs. When we assessed the expression of IRF-4 by DCs and macrophages we observed that a high percentage of DCs

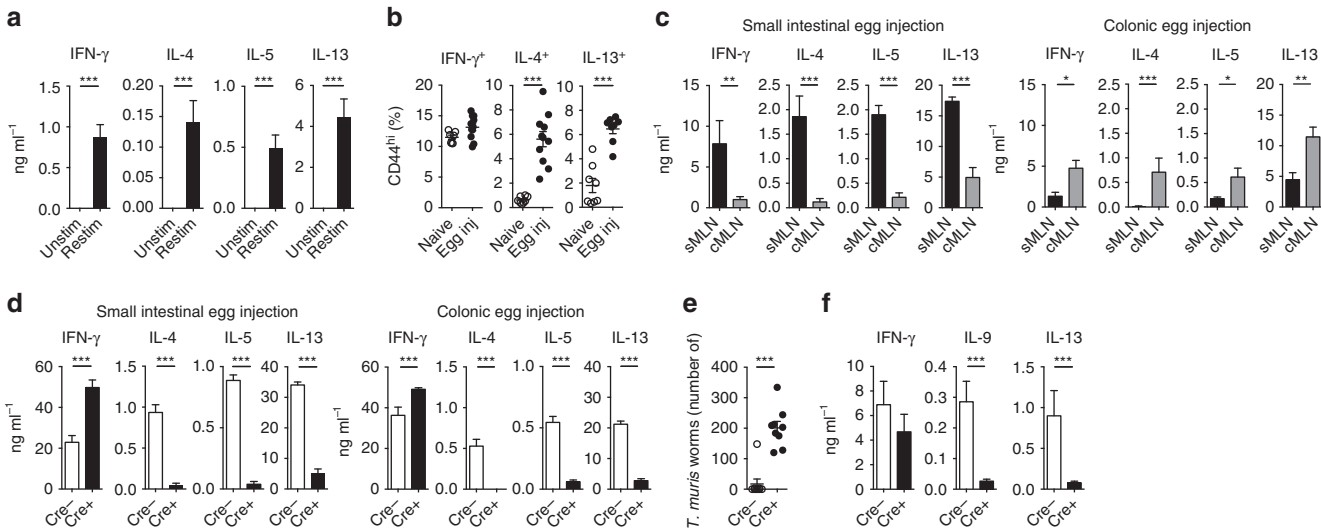

**Figure 1 | Intestinal Th2 responses to *S. mansoni* eggs and *T. muris* worms are dependent on IRF-4$^+$ CD11c$^+$ cells.** (**a**) One thousand *S. mansoni* eggs were injected into the subserosal layer of the small intestine and resulting T-cell responses were analysed after 5 days by restimulating MLN cells for an additional 3 days in the presence of SEA. Cytokines were measured from supernatants of restimulation cultures of unstimulated (Unstim) and restimulated (Restim) MLN cells (*n* = 9 mice, in three independent experiments, mean ± s.e.m., Mann–Whitney *U*-tests, ***P ≤ 0.001). (**b**) Five days after immunization with *S. mansoni* eggs (Egg inj), MLNs were harvested, CD44$^{hi}$ CD4 T cells identified by flow cytometry and levels of IFN-γ, IL-4 and IL-13 cytokine production measured after PMA/ionomycin stimulation and compared with cells harvested from naive animals (*n* = 10 mice per group, in three independent experiments, mean ± s.e.m., Mann–Whitney *U*-tests, ***P ≤ 0.001). (**c**) Cytokine responses of individually harvested small intestinal (sMLN) and colonic (cMLN) draining lymph node restimulation cultures 5 days after small intestinal (left panel) or colonic (right panel) egg injection (*n* = 9 mice per group, in three independent experiments, mean ± s.e.m., Mann–Whitney *U*-tests, *P ≤ 0.05, **P ≤ 0.01 and ***P ≤ 0.001). (**d**) Cytokine responses of restimulated MLN cells from small intestinal (left panel) or colonic (right panel) egg injected IRF-4$^{f/f}$ CD11c-cre + or littermate IRF-4$^{f/f}$ cre − BM chimeric mice (*n* = 6 mice per group, in two independent experiments, mean ± s.e.m., Mann–Whitney *U*-tests, ***P ≤ 0.001). (**e**) Worm burden in cre + or littermate cre − mice following *T. muris* infection. Mice were infected with ∼300 infectious eggs by oral gavage and worms in the colon quantified at 35 days post infection (*n* = 8–9 mice per group, in three independent experiments, mean ± s.e.m., Mann–Whitney *U*-tests, ***P ≤ 0.001). (**f**) Secreted cytokines after *T. muris* E/S antigen-specific restimulation of MLN cells 35 days after *T. muris* infection (*n* = 9 mice per group, in three independent experiments, mean ± s.e.m., Mann–Whitney *U*-tests, ***P ≤ 0.001).

expressed IRF-4, whereas only a few macrophages were IRF-4$^+$ (Supplementary Fig. 2d–g). Although macrophages expressed lower levels of IRF-4, a role for IRF-4 in these cells cannot be excluded. However, consistent with previous published work[14,18,28], we found that macrophages were absent from thoracic duct lymph of mesenteric lymphadenectomized (MLNx) mice (Supplementary Fig. 3c) and are therefore unable to prime T-cell responses in the MLNs, excluding the possibility that IRF-4$^+$ macrophages are necessary to prime Th2 responses. To further address whether IRF-4 in DCs was necessary for Th2 induction, we performed DC transfer experiments.

To verify that IRF-4$^+$ DCs were also necessary for establishing a physiological type 2 immune response against live parasites, cre + and cre − mice were infected with ∼250–300 eggs from the nematode *T. muris* by oral gavage. Under these conditions, *T. muris* infection evokes a strong Th2 response in the colon of C57BL/6 mice that mediates expulsion within 35 days[29]. We observed that cre + mice did not effectively clear adult worms by 35 days post infection, suggesting an inefficient type 2 immune response (Fig. 1e). Indeed Th2 responses were markedly decreased in cre + mice and we observed reduced production of IL-9 and IL-13 in *in vitro* MLN restimulation cultures 35 days post infection (Fig. 1f).

Thus, Th2 responses to parasite antigen present in the small intestine or colon require IRF-4-expressing CD11c$^+$ cells for their induction in the respective draining lymph nodes.

**Lymph DCs prime responses to *S. mansoni* eggs.** To determine which migratory cell populations were responsible for transporting parasite antigen from the periphery to the draining lymph nodes,

AlexaFluor660 (AF660)-labelled SEA was injected into the intestinal serosa. To directly assess migrating cell populations, thoracic duct lymph was collected from MLNx mice for 18 h after SEA-AF660 injection, using previously described techniques[14]. We observed that among all lymph migrating cells, B cells and conventional DCs labelled positive for Alexa660-labelled SEA (Fig. 2a and Supplementary Fig. 3a). DCs were the most efficient population to transport SEA-AF660 from the intestine, representing the highest proportion of AF660-labelled cells. To determine which cells were capable of inducing SEA-specific immune responses *in vivo*, fluorescence-activated cell sorting (FACS)-purified donor cells from egg-injected animals were transferred under the MLN capsule of wild-type recipient mice. We have previously used this technique to assess DC functions *in vivo*[18], inspired by an elegant study examining migration of transferred DCs into recipient lymph nodes[30].

Intranodal DC transfer allowed the direct assessment of the *in vivo* priming capabilities of the transferred cells in their physiological location. Five days after cell transfer, the injected MLNs were harvested and restimulated with SEA *in vitro* to test for antigen-specific immune responses. DCs from egg-injected donors were the only cells able to induce antigen-specific immune responses upon cell transfer, whereas B cells—despite carrying antigen—could not drive antigen-specific immune responses after transfer (Fig. 2b). In agreement with previous experiments using BM-derived DCs *in vivo*[6,31] or splenic DCs *in vitro*[32], SEA-specific Th2 induction by intestinal lymph DCs required major histocompatibility complex II (MHCII) expression (Fig. 2c and Supplementary Fig. 3b), but was independent of their ability to produce IL-4 (Supplementary Fig. 3e).

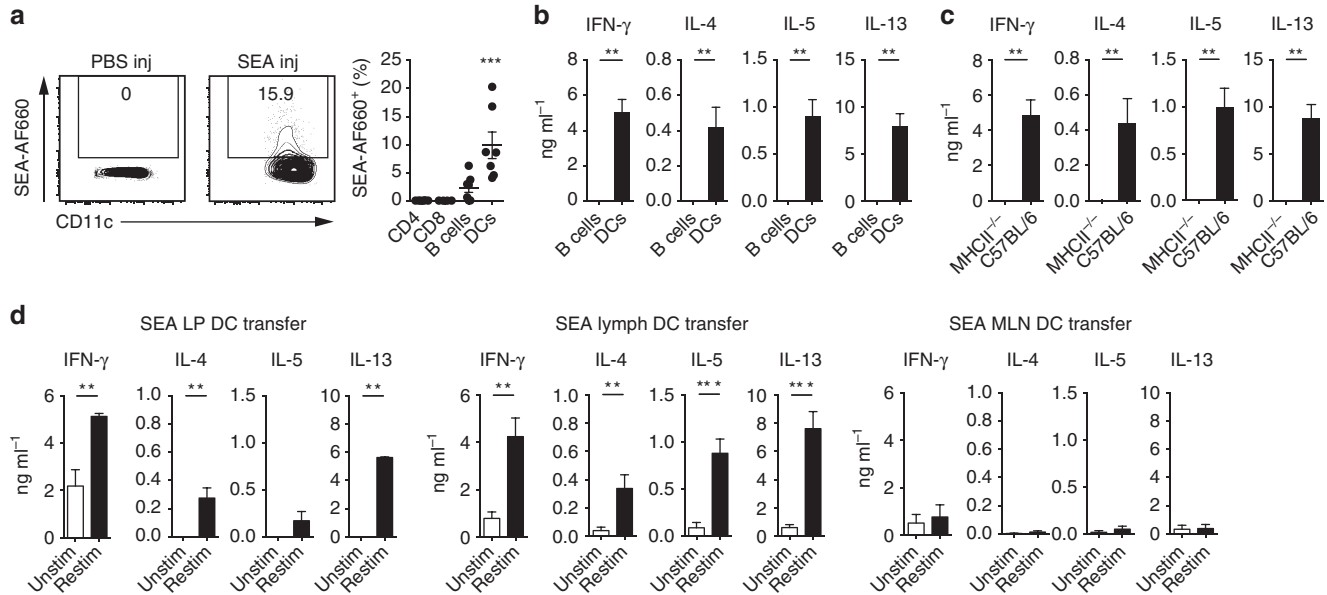

**Figure 2 | Conventional DCs drive immune responses against SEA in the MLN.** (**a**) Transport of parasite antigen was assessed by injecting AF660-labelled SEA (SEA-AF660) into the small intestine of MLNx mice and measuring the percentage of fluorescent cells in intestinal draining lymph 18 h after injection. Representative FACS plots of SEA-transporting DCs from PBS (PBS inj) and SEA (SEA inj) injected mice (left panel), and percentage of SEA-transporting lymph migrating CD4 and CD8 T cells, B cells and DCs (DCs) (right panel) are shown ($n=7$ mice in three independent experiments, mean ± s.e.m., Kruskal–Wallis test, ***$P \leq 0.001$). (**b**) Fifty thousand MHCII$^{hi}$ CD64$^-$ B220$^-$ CD11c$^{hi}$ DCs and 100,000 B cells were purified from the lymph of egg injected MLNx donor mice 18 h after injection and delivered under the MLN capsule of wild type recipient mice. After 5 days, antigen specific T-cell responses were measured in the injected MLNs by SEA restimulation for 3 days and subsequent cytokine measurement by ELISA ($n=6$ mice per group, in three independent experiments, mean ± s.e.m., Mann–Whitney $U$-tests, **$P \leq 0.01$). (**c**) Similar to **b**, CD11c$^{hi}$ CD64$^-$ B220$^-$ DCs were purified from the lymph of egg-injected C57BL/6 or MHCII$^{-/-}$ MLNx donor mice, transferred into the MLNs of recipient animals and antigen specific T-cell responses measured after *in vitro* restimulation with SEA ($n=6$ mice per group, in three independent experiments, mean ± s.e.m., Mann–Whitney $U$-tests, **$P \leq 0.01$). (**d**) Fifty thousand DCs were purified from the small intestinal LP (left panel), the lymph of MLNx mice (middle panel) or the MLNs (right panel) of C57BL/6 mice and loaded with SEA for 18 h *in vitro*. Unbound antigen was washed off and cells transferred under the MLN capsule of wild-type recipient animals. T-cell responses in the injected MLNs were measured 5 days after cell transfer by cytokine analysis of *in vitro* restimulation cultures with (Restim) or without (Unstim) SEA ($n=10$ mice per group, in three independent experiments, mean ± s.e.m., Mann–Whitney $U$-tests, **$P \leq 0.01$ and ***$P \leq 0.001$).

Antigen-specific immune responses could also be induced by DCs incubated with SEA *in vitro*. FACS-purified DCs from the intestinal lamina propria (LP), the lymph, and the MLNs of wild-type C57BL/6 animals were cultured with SEA for 18 h and transferred into recipient animals. Transferred LP-derived and lymph DCs induced antigen-specific immune responses, measured in *in vitro* restimulation cultures. However, SEA-loaded MLN DCs did not induce any antigen-specific responses after transfer (Fig. 2d).

Thus, conventional DCs are sufficient to drive *S. mansoni* egg antigen-specific immune responses in the intestinal draining lymph nodes, and process and present egg antigens to CD4 T cells in a MHCII-dependent manner.

**SEA treatment of DCs does not affect IRF-4-related genes.** To identify the intestinal DC populations driving egg antigen-specific immune responses, lymph DCs were separated by their expression of CD11b and CD103 (Supplementary Fig. 3c). Thirty thousand cells of each population were isolated from wild-type C57BL/6 MLNx lymph, incubated with SEA *in vitro* and transferred under the MLN capsule of wild-type recipient mice. Both, CD11b$^+$CD103$^-$ SP (CD11b single-positive (SP)) and CD11b$^+$CD103$^+$ double-positive (DP) DCs could induce antigen-specific Th2 and IFN-γ responses after transfer, whereas CD11b$^-$CD103$^-$ double-negative (DN) and CD11b$^-$CD103$^+$ (CD103 SP) DCs could only induce antigen-specific IFN-γ responses (Fig. 3a). Thus, only CD11b-expressing DC

populations were specialized to induce Th2 responses, whereas all DC populations could induce antigen-specific IFN-γ.

To understand the underlying molecular mechanisms that selectively enabled CD11b SP and DP DCs to prime antigen-specific Th2 responses, we performed microarray analysis and compared the gene expression profiles of sorted CD11b SP and DP DCs after *in vitro* incubation with or without SEA. Five thousand and eighteen significant loci of coding and non-coding gene elements were identified as differentially expressed between any of the four conditions (CD11b SP DCs/DP DCs, SEA-treated/untreated). Principal component analysis revealed clustering of replicate samples and a clear separation between the cell populations and between treatments (Fig. 3b). Several of the genes affected by SEA treatment have been shown to be involved in antigen presentation and T-cell differentiation, and 41 differentially expressed genes changed their expression levels more than 2-fold after SEA treatment of CD11b SP DCs. The highest fold changes were observed in downregulated MHCII-related genes, *Ccl17* and *Il1f9*, which encodes the proinflammatory cytokine IL-36γ. Costimulatory molecules such as *Cd80* and *Tnfsf4*, which encodes OX40L, and *Rasgrp3* and *Serpinb9b* were upregulated by SEA treatment of CD11b SP DCs (Fig. 3c). Thirty-three genes were differentially expressed after SEA treatment of DP DCs and a downregulation of MHCII-related genes and *Il1f9*, and an upregulation of *Rasgrp3* and *Serpinb9b* was again observed (Fig. 3c). We observed limited overlap between the differentially expressed genes with high fold

change (absolute fold change $< -2; > 2$ and analysis of variance $P$-value $< 0.05$) after SEA treatment of CD11b SP and DP DCs, which were *Rasgrp3*, *H2-Eb2* and *Il1f9* (Fig. 3d). However, none of these genes have previously been associated with Th2 cell polarization. Strikingly, IRF-4-associated genes did not change their expression profile upon treatment with SEA (Fig. 3e), despite the fact that IRF-4 expression by CD11c$^+$ cells was required for the induction of Th2 responses against *S. mansoni* eggs and *T. muris* worms.

**IRF-4 affects the migration of intestinal DCs.** Our observation that egg antigens did not induce the expression of IRF-4-associated genes in DCs suggested that the defect of Th2 responses observed in IRF-4$^{f/f}$ CD11c-cre+ mice was not due to the defective induction of these genes. We therefore investigated whether other mechanisms were involved and assessed whether IRF-4 deletion in CD11c$^+$ cells affected the number of each of the DC populations in the intestine. Consistent with reports in IRF-4$^{f/f}$ CD11c-cre+ mice[13], cre+ BM chimeras showed a

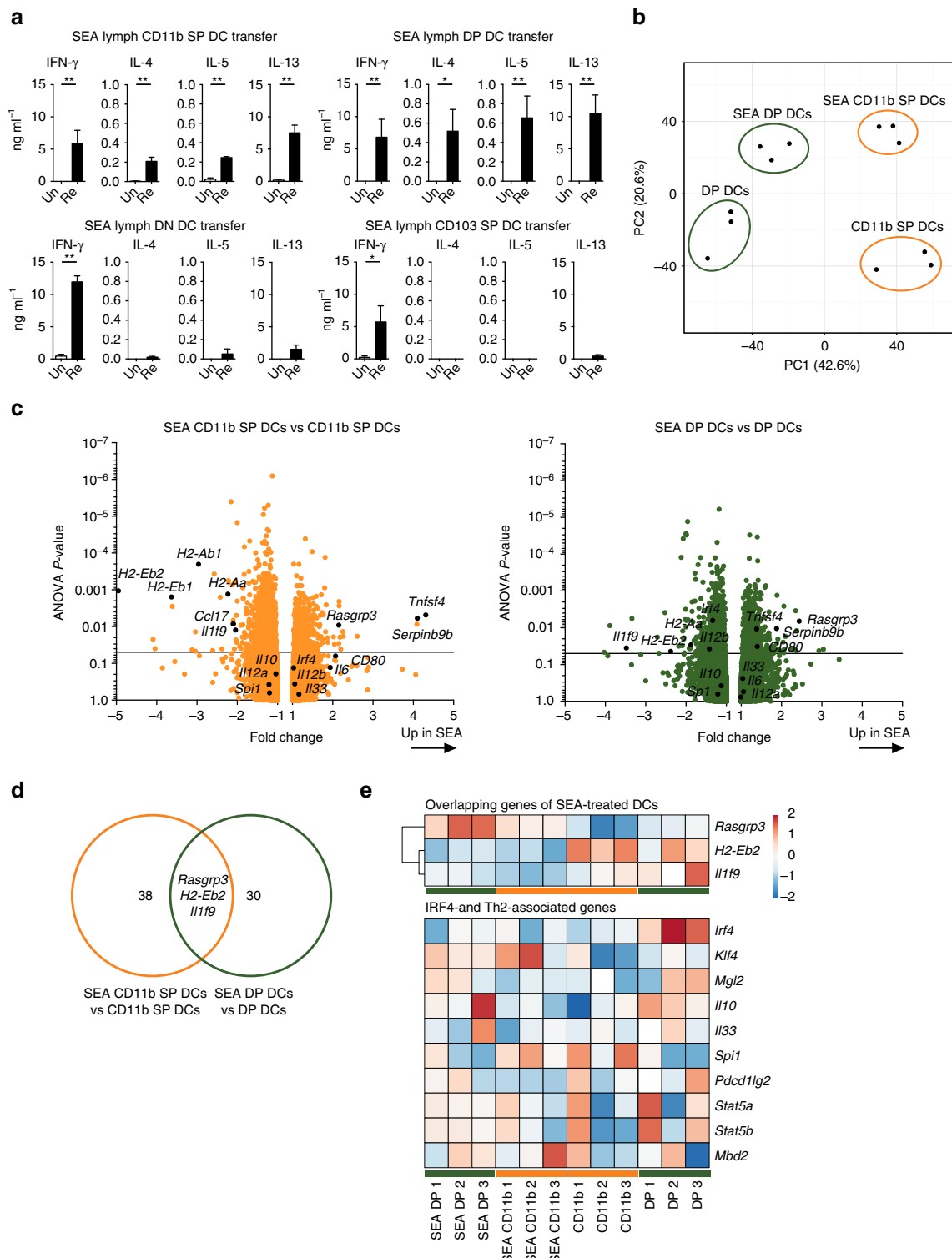

decrease in the DP DC population in the small intestine (Fig. 4a and Supplementary Fig. 4a). In the colon, a general decrease of DCs, again most strikingly observed in DP DCs, was observed (Fig. 4b). However, this developmental defect did not impair intestinal DC function. Antigen uptake, visualized by the injection of Alexa660-labelled SEA, was not affected in cre + mice and comparable numbers of intestinal LP DCs labelled positive for SEA-AF660 18 h after injection in the small intestine and colon (Fig. 4c). Furthermore, transferred SEA-loaded DP DCs from the small intestinal LP of cre + animals induced similar antigen-specific cytokine responses to C57BL/6 small intestinal LP DCs after transfer into wild-type recipient animals (Fig. 4d). Thus, IRF-4 deficiency in CD11c$^+$ cells influenced the development of intestinal LP DCs, in particular the DP DC population, and the numbers of CD11b-expressing DCs in the MLN, but did not inhibit the ability of the remaining intestinal DCs to drive Th2 cell differentiation.

To investigate whether the IRF-4-dependent reduction of LP DCs was reflected in the draining lymph nodes, we analysed DC populations in the small intestinal sMLNs and the colonic cMLNs. We observed a 50% reduction of migratory CD11b SP DCs and a near absence of migratory DP DCs in the sMLNs, as previously observed[13]. Reciprocally, the percentage of migratory CD103 SP DCs increased dramatically, but total numbers of CD103 SP DCs were not affected by the deletion of IRF-4 (Fig. 4e and Supplementary Fig. 4b). Migratory DC populations were affected to a similar extent in the cMLNs, where numbers of CD11b SP were reduced by half, DP DCs became almost absent and the numbers of CD103 SP DCs remained unaffected by the deletion of IRF-4 (Fig. 4f). Similar to LP DCs, the T-cell priming capabilities of these migratory MLN DC populations were not affected by the deletion of IRF-4. In in vitro co-cultures ovalbumin (OVA), pulsed migratory DC populations from the sMLNs and cMLNs of cre + animals drove equivalent proliferation of OVA-specific OT-II CD4$^+$ T cells compared with their cre − counterparts (Fig. 4g and Supplementary Fig. 4c). In contrast, antigen delivery to the MLNs was strongly affected in vivo. Eighteen hours after AF660-SEA injection into the small intestinal or colonic LP, the number of AF660-SEA$^+$ DCs was greatly reduced in the sMLNs and cMLNs of cre + animals (Fig. 4h). Thus, IRF-4 deficiency in CD11c$^+$ cells did not influence the capacity of DCs to prime T cells and drive Th2 differentiation, consistent with our conclusions from gene expression analysis. Rather, the striking loss of migratory CD11b SP and DP DCs from the draining lymph nodes, combined with the decrease in the amount of transported parasite antigen, were the probable cause of the inadequate intestinal Th2 responses observed in IRF-4$^{f/f}$ CD11c-cre + animals.

**Distinct DCs drive Th2 response in small intestine and colon.** As the composition of CD11b-expressing DC populations varies between the small intestine and colon[15], we assessed whether tissue-specific roles could be attributed to CD11b SP and DP DCs in priming intestinal Th2 responses. To directly assess the migration of DCs from the small intestine to the sMLNs, where priming occurs, we collected DCs from thoracic duct lymph from small intestinal MLNx (sMLNx) mice[16]. We observed that DP DCs, the most abundant DC population in the small intestine, migrated at an increased frequency, compared with PBS-injected controls, after the injection of S. mansoni eggs into the small intestine (Fig. 5a and Supplementary Fig. 5a,c). DP DCs were also the predominant population to transport small intestinally injected SEA-AF660 in sMLNx lymph (Fig. 5b). Importantly, the transfer of FACS-purified sMLNx lymph DC populations from egg-injected donor mice into recipient animals revealed that DP DCs were the only population sufficient to drive antigen-specific immune responses after transfer (Fig. 5c). This was confirmed after SEA loading of small intestinal LP DC populations, to ensure that any differences in antigen availability did not influence the results. Again, DP DCs were the most efficient population to prime antigen-specific Th2 responses against egg antigens. Similar to our previous observations, IFN-γ responses could be induced by all DC populations (Fig. 5d).

Thus, DP DCs specialize in transporting and presenting S. mansoni egg antigens from the small intestine and prime antigen-specific Th2 cells in the draining lymph nodes.

Examination of DC populations migrating from the colon revealed a different picture. In contrast to the small intestine, CD11b SP DCs migrated in increased frequency in colon-draining cMLNx lymph after the injection of S. mansoni eggs into the colon (Fig. 5e and Supplementary Fig. 5b,d). As well as being the predominant DC population within the colonic LP[15], CD11b SP DCs were the only population to carry fluorescently labelled SEA-AF660 in the lymph after colonic injection (Fig. 5f). Furthermore, colonic CD11b SP LP DCs were the most efficient at inducing Th2 responses against in vitro-loaded SEA after transfer (Fig. 5g).

Thus, in contrast to the small intestine, CD11b SP DCs were responsible for transporting and presenting S. mansoni egg antigens from the colon and were the most efficient population for priming antigen-specific Th2 cell in the colon-draining lymph nodes.

---

**Figure 3 | CD11b$^+$ DCs drive antigen-specific Th2 responses but SEA does not alter IRF-4-related gene expression.** (**a**) Thirty thousand cells of each of the four intestinal DC populations, distinguished by their expression of CD11b and CD103, were purified from the MLNx C57BL/6 animals and incubated with SEA for 18 h in vitro. Unbound antigen was washed off and cells transferred under the MLN capsule of wild-type recipient animals. T-cell responses in the injected MLNs were measured 5 days after cell transfer by cytokine analysis of in vitro restimulation cultures with (Re) or without (Un) SEA ($n = 6$ mice per group, in three independent experiments, mean ± s.e.m., Mann–Whitney $U$-tests, **$P \leq 0.01$ and ***$P \leq 0.001$). (**b**) The gene expression profiles of SEA-treated or -untreated CD11b$^+$CD103$^-$ single-positive (CD11b SP) and CD11b$^+$CD103$^+$ double-positive (DP) DCs were analysed by microarray analysis. Principle component analysis of the 5,018 significant loci of coding and non-coding gene elements identified to be differentially expressed between any of the four conditions ($n = 3$ samples per condition, unpaired one-way (single factor) analysis of variance (ANOVA) for each pair of condition groups, ANOVA $P$-value (condition pair) < 0.05). (**c**) All gene loci from CD11b SP (left panel) and DP DCs (right panel) were compared between SEA-treated and -untreated cells and the absolute fold change and ANOVA $P$-value visualized using volcano plots. Coding genes of interests are highlighted ($n = 3$ samples per condition, unpaired one-way (single factor) ANOVA for each pair of condition groups). (**d**) All coding genes that were found to be differentially expressed within each condition pair with absolute fold change < − 2; > 2 and ANOVA $P$-value < 0.05 were selected and summarized. Genes that changed in both CD11b SP and DP DCs are highlighted ($n = 3$ samples per condition, unpaired one-way (single factor) ANOVA for each pair of condition groups; ANOVA $P$-value (condition pair) < 0.05). (**e**) The relative expression intensities of the overlapping genes from **d** and of IRF-4- and Th2-associated genes determined from the literature are shown for each individual sample (changes of overlapping genes are significant, whereas changes of IRF-4- and Th2-associated genes are not significant) ($n = 3$ samples per condition, unpaired one-way (single factor) ANOVA for each pair of condition groups for the two condition groups; ANOVA $P$-value (condition pair) < 0.05).

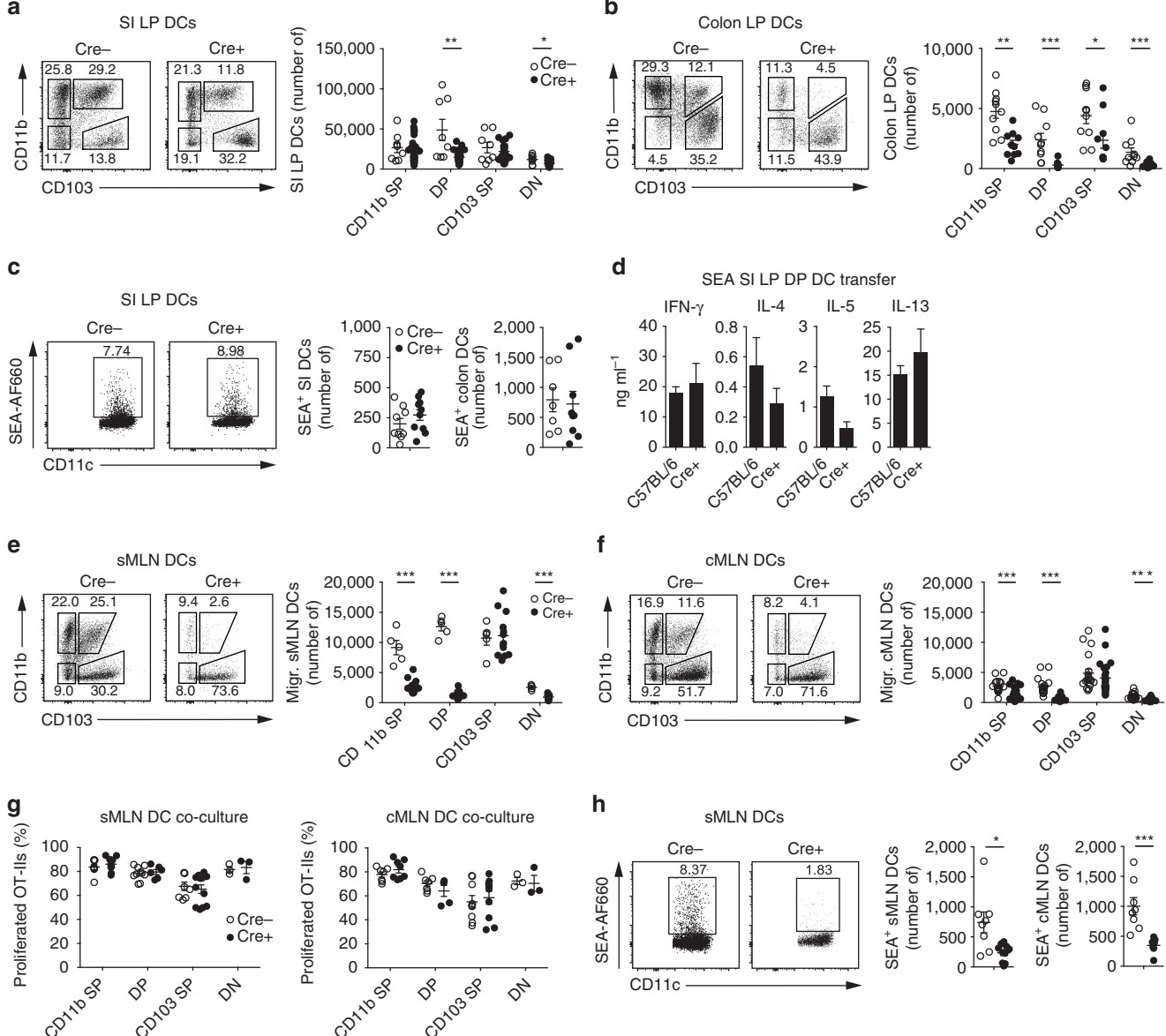

**Figure 4 | IRF-4 deficiency alters DC population composition to reduce antigen availability in the MLNs.** (**a**) Representative percentages and total numbers of CD11b and CD103-expressing DC populations from the small intestines of IRF-4$^{f/f}$ CD11c-cre + or littermate IRF-4$^{f/f}$ cre − ) bone-marrow (BM) chimeras (n = 8 mice per group, in three independent experiments, mean ± s.e.m., Mann–Whitney U-tests, *P ≤ 0.05 and **P ≤ 0.01).
(**b**) Representative percentages and total numbers of DC populations from the colon of cre + and cre − BM chimeras (n = 8 mice per group, in three independent experiments, mean ± s.e.m., Mann–Whitney U-tests, *P ≤ 0.05, **P ≤ 0.01 and ***P ≤ 0.001). (**c**) Uptake of AF660 labelled SEA (SEA-AF660) by intestinal DCs from cre + and cre − BM chimeras 18 h after injection into the small intestine or colon. Representative FACS plots from the small intestine (left panel) and total numbers of SEA-AF660 $^+$ small intestinal (middle panel) and colonic (right panel) DCs are shown (n = 7–10 mice per group, in three independent experiments, mean ± s.e.m., Mann–Whitney U-tests (not significant)). (**d**) Thirty-thousand CD11b $^+$ CD103 $^+$ double-positive (DP) DCs from the small intestine of C57BL/6 or cre + BM chimeras were incubated with SEA for 18 h in vitro and transferred under the MLN capsule of wild-type recipient animals. Antigen specific T-cell responses in the injected MLNs were measured 5 days after cell transfer by cytokine analysis of in vitro SEA restimulation cultures (n = 6 mice per group, in three independent experiments, mean ± s.e.m., Mann–Whitney U-tests (P(IFN-γ) = 0.59, P(IL-4) = 0.82, P(IL-5) = 0.09 and P(IL-13) = 0.50). (**e**) Representative percentages and total numbers of DC populations from the sMLNs of cre + and cre − BM chimeras (n = 5–12 mice per group, in three independent experiments, mean ± s.e.m., Mann–Whitney U-tests, ***P ≤ 0.001). (**f**) Representative percentages and total numbers of DC populations from the colonic draining MLNs (cMLNs) of cre + and cre − BM chimeras (n = 12 mice per group, in three independent experiments, mean ± s.e.m., Mann–Whitney U-tests, ***P ≤ 0.001). (**g**) Six thousand sMLN (left panel) or 3,000 cMLN (right panel) DC populations from cre + and cre − BM chimeras were pulsed with OVA and co-cultured with CFSE-labelled OT-II MLN cells for 3 days. In vitro OT-II CD4 T-cell proliferation was assessed by CFSE dilution and compared between cre + and cre − DC populations (n = 3–16 duplicate cocultures per group, in three independent experiments, mean ± s.e.m., Mann–Whitney U-tests (not significant)). (**h**) Transport of SEA-AF660 by cre + and cre − migratory DCs to the draining lymph nodes 18 h after injection into the small intestine or colon. Representative FACS plots from the sMLN (left panel) and total numbers of SEA-AF660 $^+$ migratory sMLN (middle panel) and cMLN (right panel) DCs are shown (n = 8 mice per group, in three independent experiments, mean ± s.e.m., Mann–Whitney U-tests, *P ≤ 0.05 and ***P ≤ 0.001).

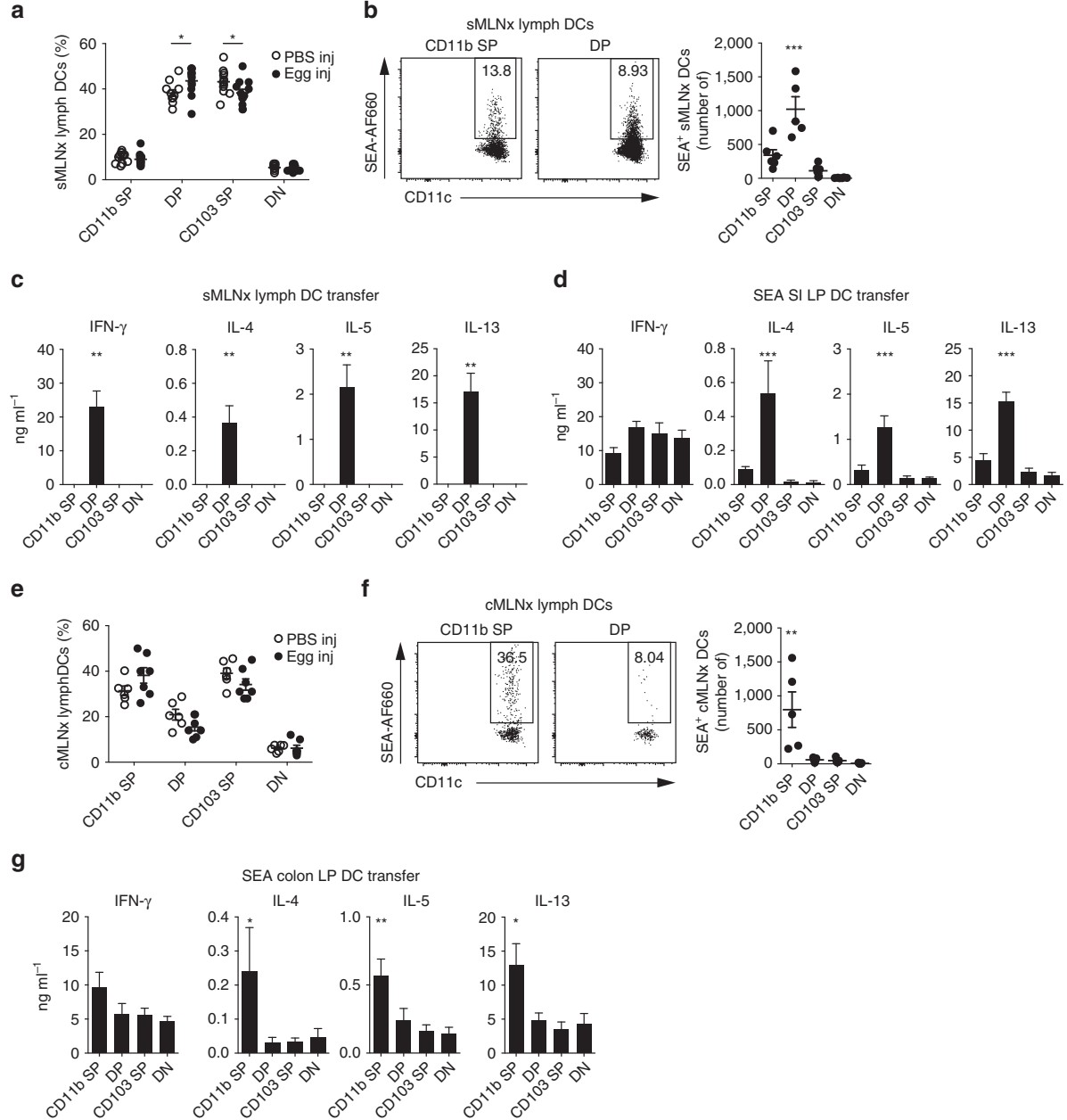

**Figure 5 | Th2 responses in the small intestine and colon are driven by distinct CD11b⁺ DCs.** (**a**) Frequency of lymph migrating DC populations 18 h after the injection of PBS (PBS inj) or *S. mansoni* eggs (Egg inj) into the small intestine of sMLNx C57BL/6 animals (*n* = 10–11 mice per group, in three independent experiments, mean ± s.e.m., Mann–Whitney *U* tests (*P*(CD11b SP) = 0.25, *P*(DP) = 0.02, *P*(CD103 SP) = 0.04 and *P*(DN) = 0.14). (**b**) Transport of AF660-labelled SEA (SEA-AF660) by lymph-migrating DC populations 18 h after small intestinal injection into sMLNx animals. Representative FACS plots from CD11b⁺CD103⁻ single-positive (CD11b SP) and CD11b⁺CD103⁺ double-positive DCs (left panel) and total numbers of SEA-AF660⁺ DC populations (right panel) are shown (*n* = 5 mice per group in two independent experiments, mean ± s.e.m., Kruskal–Wallis test, ***P* ≤ 0.001). (**c**) Thirty-thousand DCs from each population were isolated from the lymph of small intestinal egg-injected sMLNx mice and transferred under the MLN capsule of wild-type recipient animals. Antigen-specific T-cell responses in the injected MLNs were measured 5 days after cell transfer by cytokine analysis of *in vitro* SEA restimulation cultures (*n* = 6 mice per group, in three independent experiments, mean ± s.e.m., Kruskal–Wallis test, **P* ≤ 0.01). (**d**) Thirty-thousand DCs from each population were isolated from the small intestine of C57BL/6 mice, incubated with SEA *in vitro* and transferred under the MLN capsule of wild-type recipient animals. Antigen-specific immune responses were analysed as in **c** (*n* = 6 mice per group, in three independent experiments, mean ± s.e.m., Kruskal–Wallis test, ***P* ≤ 0.001). (**e**) Frequency of lymph-migrating DC populations 18 h after the injection of PBS or *S. mansoni* eggs into the colon of colonic MLNx (cMLNx) C57BL/6 animals (*n* = 6-7 mice per group, in three independent experiments, mean ± s.e.m., Mann–Whitney *U*-tests (*P*(CD11b SP) = 0.17, *P*(DP) = 0.17, *P*(CD103 SP) = 0.22 and *P*(DN) = 0.71). (**f**) Transport of SEA-AF660 by lymph-migrating DC populations 18 h after injection into the colon of cMLNx animals. Representative FACS plots from CD11b SP and DP DCs (left panel), and total numbers of SEA-AF660⁺ DC populations (right panel) are shown (*n* = 5 mice per group, in two independent experiments, mean ± s.e.m., Kruskal–Wallis test, ***P* ≤ 0.01). (**g**) Thirty-thousand DCs from each population were isolated from the colon of C57BL/6 mice, incubated with SEA *in vitro* and transferred under the MLN capsule of wild-type recipient animals. Antigen-specific immune responses in the injected MLNs were measured 5 days after cell transfer by cytokine analysis of *in vitro* SEA restimulation cultures (*n* = 6 mice per group, in three independent experiments, mean ± s.e.m., Kruskal–Wallis test, **P* ≤ 0.05 and ***P* ≤ 0.01).

## Discussion

Many cell types have been implicated in inducing type 2 immune responses against *S. mansoni* egg antigens, including monocyte-derived DCs[7], conventional DCs[11] and basophils[33]. In this context, the intestine is an important tissue, being heavily affected by penetrating *S. mansoni* eggs during live infection, and the target of many Th2-inducing helminth parasite infections[34]. To investigate which cell populations are sufficient to induce intestinal Th2 responses, we establish an experimental immunization procedure for controlled delivery of eggs into intestinal subserosal tissue. We observe that egg antigen-specific CD4 T-cell responses are induced in the draining MLNs (Fig. 1), which, as previously reported[11,35,36], produced both IFN-γ- and the Th2-associated cytokines IL-4, IL-5 and IL-13. *S. mansoni* eggs induce antigen-specific immune responses in both the small intestinal and colonic draining lymph nodes after injection into the respective tissues. IRF-4-expressing CD11c[+] cells are critical for the induction of Th2 responses both in the small intestine and colon (Fig. 1), as Th2 immune responses did not develop in IRF-4[f/f] CD11c-cre+ mice, similar to previous reports in the lung[21] and the small intestine during *Nippostrongylus brasiliensis* infection[20].

As CD11c is expressed by monocyte-derived DCs, conventional DCs and macrophages in the intestine[37], we address which specific cell population transports egg antigen from the intestine to the draining lymph nodes and is sufficient to drive antigen-specific immune responses. Injection of fluorescently labelled SEA enables identification of B cells and conventional DCs transporting egg antigens to the MLN. Transfer of cells directly under the MLN capsule allows for delivery of cells by their physiological route of entry and to assess their priming capabilities *in vivo*[18,30]. MHCII-expressing DCs, but not B cells, from egg-injected donor mice are sufficient to induce egg antigen-specific immune responses in the recipient MLN (Fig. 2). Furthermore, both intestinal LP DCs and lymph DCs isolated from wild-type C57BL/6 mice can present SEA after *in vitro* incubation. Thus, our data indicate that conventional DCs transport parasite antigen from the intestine to the draining lymph nodes and are sufficient to directly prime antigen-specific immune responses. Further analysis of the four intestinal DC populations that can be defined by the expression of CD11b and CD103 (refs 12–14) reveals that SEA-loaded CD11b[+]CD103[−] single-positive (CD11b SP) and CD11b[+]CD103[+] double-positive (DP) DCs are specialized to prime antigen-specific Th2 and IFN-γ responses upon transfer, whereas CD11b[−]CD103[−] double-negative (DN) and CD11b[−]CD103[+] single-positive (CD103 SP) DCs can only induce IFN-γ responses (Fig. 3). This observation is in line with previous findings that demonstrate that CD11b-expressing DCs from the spleen, lung and skin induce Th2 responses[9,31,38–40]. In contrast, CD103 SP DCs have been shown to negatively regulate Th2 responses, by their constitutive expression of IL-12 (ref. 41). The balance between the different DC populations is therefore critical for determining the outcome of the T-cell response.

Microarray analysis of the Th2-priming CD11b-expressing DCs reveals that the expression levels of genes involved in antigen presentation and T-cell differentiation are influenced by incubating CD11b SP and DP DCs with SEA. For example, *CD80* and *Tnfsf4*, which encodes OX40L, are upregulated by SEA. These costimulatory markers are required for T-cell differentiation and OX40L has been previously been shown to be induced by SEA[42]. Furthermore, the downregulation of MHCII-related genes, proinflammatory mediators such as IL-36γ, encoded by *Il1f9*, and the chemokine CCL17, which has been shown to recruit proinflammatory Th2 cells during allergy[43] could suggest that SEA dampens proinflammatory responses, as previously observed *in vitro* and *in vivo*[31,32,44].

We anticipated that our expression analysis would reveal genes induced in both CD11b-expressing DC populations that contribute to their common ability to prime Th2 responses. However, we observe little overlap and only three genes are differentially expressed in SEA-cultured CD11b SP and DP DCs. *Rasgrp3* is upregulated, whereas *H2-Eb2* and *Il1f9* are downregulated upon SEA incubation in both populations. RAS activators, such as RasGRP3, provide a key link between cell surface receptors and RAS activation, and RasGRP3 has been shown to control CCR9-dependent entry of early thymic progenitors in the thymus[45], B-cell receptor signalling[46] and the production of Toll-like receptor (TLR)-triggered proinflammatory cytokines in macrophages[47]. Together with the downregulation of *H2-Eb2* and *Il1f9*, a dampening of proinflammatory responses is thus observed in these SEA-cultured DCs. However, no direct involvement of these genes in type 2 immune response has previously been reported and comparison with data sets from Th2-impaired Mbd2[−/−] bone-marrow DCs[35] reveal no commonality. Surprisingly, the expression of IRF-4 and its binding partner PU.1, encoded by *Spi1*, are not upregulated by SEA, despite the fact that IRF-4 expression by CD11c[+] cells is required to drive Th2 polarization. These genes have been shown to promote the expression of OX40L[48], IL-10 and IL-33 (ref. 21), which influence Th2 polarization, but are not upregulated in DCs after SEA treatment. Other genes that have been implicated in polarizing and enhancing Th2 responses, such as *Mgl2* (refs 20,49), *Pdcd1lg2* (refs 50,51), which encodes PDL2, *Stat5a/b*[52] and *Mbd2* (ref. 35) are also not upregulated by SEA (Fig. 3), leading us to conclude that *S. mansoni* egg antigens induce the ability to drive Th2 polarization in intestinal DCs by mechanisms that may not be revealed by gene expression analysis.

A key element of our work is that we have discovered that IRF-4 deficiency does not directly interfere with intestinal DC ability to induce T-cell priming or drive Th2 differentiation (Fig. 4), despite being previously suggested in the literature[21,53]. Rather, IRF-4 deficiency limits the number of intestine-derived CD11b-expressing DCs in the draining lymph nodes, consistent with previous observations in the intestine[13] and the skin[12]. Impaired survival[12,13] and lack of migration[54] have both been suggested to cause this pronounced IRF-4-dependent decrease in CD11b-expressing DCs. As a result, limited amounts of egg antigen are present in the MLNs of IRF-4[f/f] CD11c-cre+ mice, which probably causes the impaired Th2 responses in these mice (Fig. 4). Thus, IRF-4 does not directly control the ability of intestinal DCs to polarize Th2 cells, but rather affects the number of CD11b-expressing DCs, probably by influencing their differentiation and survival, and thus the amount of presented antigen in the draining lymph nodes.

As the composition of CD11b-expressing DCs varies along the gastrointestinal tract, we assessed whether CD11b SP and DP DC populations play tissue-specific roles. We and others have observed that DP DCs are the most abundant DCs in the small intestine, whereas CD11b SP DCs are more frequent in the colon[15,16]. These differences can be observed within the lamina propria (LP), as well as in tissue-specific draining lymph, and the respective draining lymph nodes[16]. We observe that after small intestinal egg injection DP DCs are present at increased frequency within lymph, suggesting increased migration (Fig. 5). It is well established that TLR activation can lead to DC migration in the intestine, which has been shown for R848 (ref. 55), a TLR7/8 agonist, and soluble flagellin[3,56], which activates TLR5. DCs do not require MyD88-mediated TLR signalling to induce Th2 responses against *S. mansoni* eggs[57], but the effects of *S. mansoni* eggs on DC migration have not previously been addressed. Diverse pathogen-associated molecular patterns, such as Omega-1, have been identified in *S. mansoni* eggs and SEA[58,59],

indicating that a range of parasite molecules may influence DC migration.

As well as displaying increased migration, DP DCs are the most numerous population to carry fluorescently labelled SEA from the small intestine to the draining lymph nodes. Transfer of purified DC populations from egg-injected donor mice reveals that DP DCs are also the only DCs capable of inducing antigen-specific immune responses in recipient animals (Fig. 5). This is likely to be both due to their ability to capture egg antigen but also an intrinsic specialisation for inducing Th2 responses, which we verified by transfer of *in vitro* SEA-loaded small intestinal LP DCs. In this system, all populations can induce antigen-specific IFN-γ responses but only DP DCs induce strong Th2 responses, suggesting that DP DCs from the small intestine are either specialized to process and present specific Th2-inducing antigens within SEA or, more probably, can generate specific signals that drive the differentiation of Th2 cells. In contrast to the small intestine, CD11b SP DCs migrate in increased frequency in lymph of egg-injected animals after colonic egg injection, transport fluorescently labelled SEA from the colon and are the predominant population to induce Th2 responses when loaded with SEA (Fig. 5). Unexpectedly therefore, the Th2-inducing DC populations in the small intestine and colon are different, revealing unappreciated complexity in the functional specialisations of DCs in different tissues.

We thus demonstrate that specific populations of intestinal CD11b-expressing DCs are specialized to prime Th2 cells in the small intestine and colon, and speculate that this capacity extends beyond antigens from *S. mansoni* eggs and may also be relevant for other parasitic antigens or intestinal food allergens. The differences in the functions of CD11b-expressing DC populations between the small intestine and colon also provide clear evidence that the induction of immune responses in these tissues is controlled differently. This idea is supported by recent findings that have demonstrated that oral tolerance in the small intestine and colonic tolerance are driven by distinct populations of tissue-specific DCs[60]. Beyond advancing our understanding of the immunological differences between these tissues, this raises the possibility that diseases in the small intestine and colon could also be influenced by distinct populations of DCs. Many important infections and inflammatory conditions (for example, parasite infections, Crohn's disease, ulcerative colitis and celiac disease) selectively affect the small intestine or colon and DCs have been shown to contribute to disease induction or progression. Delineation of the roles of specific DC populations in these conditions could thus reveal novel pathways that may be precisely and independently targeted to beneficially modify the involved protective or pathogenic immune responses. Thus, our identification of the tissue-specific DC populations that induce Th2 responses against *S. mansoni* eggs in the intestine reveals novel insight into the induction of intestinal type 2 immune responses. It also impacts our understanding of intestinal immune responses in general, by demonstrating that different tissue-specific DC populations are responsible for driving similar responses in anatomically distinct intestinal locations.

## Methods

**Mice.** C57BL/6 (C57BL/6JOIaHsd) were ordered from Envigo and C57BL/6.SJL, IL-4$^{-/-}$, OT-II, IRF-4$^{f/f}$ CD11c-cre and IRF-4$^{f/f}$ cre− mice (all on C57BL/6 background) were bred and housed under specific pathogen free conditions at the University of Glasgow, UK, or at Lund University, Sweden. Age- and gender-matched adult animals were used in each individual experiment, which were approved by the University of Glasgow Animal Welfare Ethical Review Board and the Malmö/Lund Ethical board for Animal research and performed under licenses issued by the UK Home Office and the Swedish Board of Agriculture. IRF-4$^{f/f}$ CD11c-cre and IRF-4$^{f/f}$ cre− bone-marrow (BM) chimeras were created by lethally irradiating 6-week-old C57BL/6.SJL recipients with 10 gray using a Small Animal Radiation Research Platform (Xstrahl) and reconstituted with 2–4 × 10$^6$ IRF-4$^{f/f}$ CD11c-cre + or IRF-4$^{f/f}$ cre − BM cells. Experiments with irradiated animals were performed 8–10 weeks after irradiation.

**Surgical procedures.** All surgical procedures were carried out under aseptic conditions and inhalation anaesthesia with Isoflurane (Abbot Animal Health). For egg injections, 1,000 freeze/thawed *S. mansoni* eggs were resuspended in 20 µl DPBS (Life Technologies) and injected into the intestinal LP of anaesthetized mice. *S. mansoni* eggs for these studies were isolated under sterile conditions from the livers of infected C57BL/6 mice before cryopreservation and SEA was prepared by homogenization and ultracentrifugation of eggs, and concentrated by vacuum dialysis to 1 mg ml$^{-1}$ in DPBS[61]. For subcapsular injections, 6-week-old male mice were fed 0.2 ml olive oil to visualize the MLN capsule and MLNs were accessed by laparotomy. Cells were resuspended in 5 µl DPBS and injected under the MLN capsule. MLNx was performed on 6-week-old male mice by laparotomy and blunt dissection of the small intestinal or colonic draining lymph nodes. After 6 weeks, MLNx mice were fed 0.2 ml olive oil to visualize the lymphatics and the thoracic lymph duct was accessed by laparotomy and cannulated by the insertion of a polyurethane medical grade intravascular tube (2Fr; Linton Instrumentation). Lymph was collected for 18 h on ice in DPBS supplemented with 20 U ml$^{-1}$ of heparin sodium (Wockhardt UK).

**Infection models.** Mice were infected with ∼250–300 infective *T. muris* eggs from the E (Edinburgh) isolate by oral gavage. Adult worms were isolated from the colons of infected C57BL/6 mice. For the preparation of eggs and parasite E/S antigens worms were cultured in sterile RPMI 1640 supplemented with 500 U ml$^{-1}$ penicillin and 500 µg ml$^{-1}$ streptomycin (all Thermo-Fisher Scientific) and incubated at 37 °C for 24 h. Eggs were collected by centrifugation and parasite antigens concentrated using centriprep-centrifugal columns with 10,000 NMWL (Merck-Millipore) and dialysed to DPBS using Amicon Ultracel-3 K Centrifugal Filters with 3000 NMWL (Merck-Millipore) to a final concentration of 1 mg ml$^{-1}$. To assess worm burden, colons were isolated and frozen at − 20 °C; tissues were subsequently thawed and worms scraped free from the tissue and counted under a microscope.

**Cell isolation.** MLNs were digested using RPMI 1640 (Life Technologies) supplemented with 8 U ml$^{-1}$ Liberase and 10 µg ml$^{-1}$ DNase (all Sigma-Aldrich) for 45 min at 37 °C in a shaking incubator and single-cell suspensions were prepared using a 40 µm cell strainer (Greiner Bio One). Intestines were excised, cleaned and cut into 0.5 cm segments. Segments were washed in HBSS (Life Technologies) supplemented with 2 mM EDTA (Sigma-Aldrich) twice for 15 min at 37 °C in a shaking incubator. Small intestinal segments were digested in RPMI 1640 supplemented with 1 mg ml$^{-1}$ Collagenase VIII (Sigma-Aldrich) and 10% FCS for 15 min at 37 °C in a shaking incubator. Colons were digested with RPMI supplemented with 0.425 mg ml$^{-1}$ Collagenase V (Sigma-Aldrich), 0.425 mg ml$^{-1}$ Collagenase D (Roche), 1 mg ml$^{-1}$ Dispase (Gibco), 30 µg ml$^{-1}$ DNase (Roche) and 10% FCS for 40 min at 37 °C in a shaking incubator. Single-cell suspensions were prepared using a 100 and 40 µm cell strainer (Corning). Lymph cells were passed through a 40 µm cell strainer (Greiner Bio One) and stained directly.

***In vitro* restimulation cultures and cytokine measurement.** MLN cells (1 × 10$^6$) were cultured in X-vivo 15 media (Lonza) supplemented with 1% L-glutamine (Invitrogen), 0.1% 2-mercaptoethanol (Sigma-Aldrich) and 15 µg ml$^{-1}$ SEA in round bottom 96-well plates (Corning) at 37 °C and 5% CO$_2$. Supernatants were collected after three days and cytokines detected using the IL-4 (88-7044-77), IL-5 (88-7054-77), IL-13 (88-7137-77), IL-17 (88-7371-77) and IFN-γ (88-7341-77) 'ready-set-go' ELISA kits (eBioscience) following the manufacturer's instructions. For *T. muris* E/S-antigen-specific restimulation of total MLN cells, 0.5 × 10$^6$ cells were cultured with 50 µg ml$^{-1}$ E/S antigens for 48 h at 37 °C and 5% CO$_2$. Supernatants were collected and frozen at − 20 °C before cytokine analysis. Cytokine concentrations were determined by cytometric bead array kit (BD Biosciences) according to the manufacturer's instructions. Samples were acquired on a BD LSR II flow cytometer (BD Biosciences) and data analysed with FCAP array v3.0 software.

**Antibodies for flow cytometric analysis and cell sorting.** Mouse tissue cell surface markers and intracellular cytokines were stained using combinations of fluorescently labelled primary antibodies at a dilution of 1:200. These included anti-CD4 (clones GK1.5 and RM4-5), anti-CD8a (53–6.7), anti-CD44 (IM7), anti-CD45R/B220 (RA3-6B2), anti-CD11c (N418), anti-I-A/I-E (M5/114.15.2), anti-CD11b (M1/70), anti-CD103 (2E7), anti-CD64 (X54-5/7.1), anti-Ly6C (HK1.4), anti-TCR Vα2 (B20.1), anti-IL4 (11B11), anti-CD45 (30-F11), anti-CD45.1 (A20) and anti-CD45.2 (104) purchased from Biolegend, and anti-IFNγ (XMG1.2), anti-IRF-4 (3E4), anti-GATA3 (TWAJ) and anti-IL13 (eBio13A) from eBioscience. SEA was fluorescently labelled using the Microscale Antibody Labelling Kit (Life Technologies) following the manufacturer's instructions. For intracellular transcription factor staining, cells were fixed and permeabilized using the eBioscience Foxp3/Transcription Factor Staining Buffer

Set and intracellular staining was performed following the manufacturer's instructions. Cells were analysed using a LSRII flow cytometer running FACSDiva Software (BD Bioscience) and analysed using FlowJo Software (Tree Star). 7AAD (Biolegend) or Fixable Viability Dye eFluor780 (eBioscience) were used to exclude dead cells from analysis. For cell sorting, DCs were gated on by selecting live $CD45^+$ $CD45R/B220^-$ $CD64^-$ $Ly6C^-$ $CD11c^{hi}$ $MHCII^{hi}$ single cells and individual CD11b/CD103-expressing populations were sorted using an AriaIII cell sorter (BD Bioscience). Cells undergoing subsequent antigen loading were incubated in supplemented (as above) X-vivo 15 media (Lonza) with $15\,\mu g\,ml^{-1}$ SEA for 18 h or $2\,mg\,ml^{-1}$ OVA protein (Sigma-Aldrich) for 2 h at 37 °C and 5% $CO_2$.

**In vitro cell stimulation.** For intracellular staining experiments $2\times10^6$ MLN cells were incubated in RPMI 1640 supplemented with $2.5\,ng\,ml^{-1}$ PMA (Sigma-Aldrich), $1\,\mu g\,ml^{-1}$ ionomycin (Invitrogen), 0.5% GolgiStop (BD Bioscience) and 10% FCS for 4 h at 37 °C, after which cell surface markers were stained. Cells were fixed and permeabilized using the eBioscience Foxp3/Transcription Factor Staining Buffer Set (eBioscience) and intracellular staining was performed following the manufacturer's instructions.

**In vitro co-cultures.** For in vitro OT-II co-cultures $2\times10^5$ OT-II MLN cells were labelled with CFSE (eBioscience) at a dilution of 1:1,000 and cocultured with 6,000 sMLN or 3,000 cMLN FACS-sorted DCs from each population. Each population had been pulsed with $2\,mg\,ml^{-1}$ of OVA (Worthington, Lakewood) for 2 h at 37 °C and 5% $CO_2$ and then extensively washed. After 3 days of co-culture, cells were stained for flow cytometry and CFSE dilution was assessed to quantify cell proliferation.

**Microarray analysis.** Total RNA from SEA-incubated lymph DC populations (as above) was isolated with the RNeasy micro kit (Qiagen) and prepared for microarray analysis at Hologic Ltd using the Affymetrix Mouse Transcriptome Pico Assay 1.0. RNA samples were applied to a Mouse Transcriptome Array 1.0 (Affymetrix). Scanned CEL files were background corrected, normalized and summarized by using the Affymetrix Expression Console Software 1.4. Differential gene expression was analysed using the Affymetrix Transcription Analysis Console Software 3.0 and visualized using Prism 6 Software (GraphPad). Principal component analysis analysis and heatmap visualization were conducted using ClustVis[62].

**Statistical analysis.** Based on analyses of preliminary experiments, group sizes were chosen to ensure that a twofold difference between means, where the common standard deviation was less than or equal to half of the smaller mean, could be detected with a power of at least 80%. Prism 6 Software (GraphPad) was used to calculate the s.e.m. and statistical differences between groups were calculated using Mann–Whitney U-tests and Kruskal–Wallis tests, where appropriate, with $P<0.05$ being considered as significant.

**Data availability.** Microarray gene expression data are available from the Gene Expression Omnibus, accession number GSE91381. All other relevant data are available from the authors.

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

## Acknowledgements

We thank V. Cerovic, S. Houston, A. Schridde, A. Ivens and L. Webb for their guidance and advice, and the staff of the Central Research Facility and the Institute of Infection, Immunity and Inflammation Flow Cytometry Facility at the University of Glasgow for their assistance. We also thank S. Brown for technical support in generation of *S. mansoni* eggs and SEA, using infected snails that were supplied by the National Institute of Allergy and Infectious Diseases Schistosomiasis Resource Center at the Biomedical Research Institute (Rockville, MD), through Contract HHSN272201000005I distributed through BEI Resources, with the help of M. Wilson (The Francis Crick Institute). This work was supported by grants from the Wellcome Trust to J.U.M. and S.W.M. (099784/Z/12/Z), and by a grant from the Medical Research Council (MR/K021095/1) to S.W.M. A.S.M. is part of the MCCIR, which is a joint venture between the University of Manchester, GSK and AstraZeneca. This work was also supported by a Sapere Aude III senior research grant from the Danish research council, the Lundbeckfonden (grant number R155-2014-4184) and grants from the Swedish research council to W.W.A. and grants from the Swedish Research Council, and the Crafoord, Carl Trygger and Per Eric & Ulla Schyberg Foundations to M.S.-F.

## Author contributions

J.U.M. and M.D. performed the experiments. W.W.A., A.S.M. and M.S.-F. provided reagents and with S.W.M. helped direct the study. J.U.M. conceived the project with S.W.M. and A.S.M., and wrote the manuscript with S.W.M.

## Additional information

**Competing interests:** The authors declare no competing financial interests.

