## [Peer Review File · Nature Communications]

Reviewers' comments:

Reviewer #1 (Helminth, innate) (Remarks to the Author):

In this ms the authors have employed both an elegant and novel approach and do report insightful observations.

The ms describes a well-performed, technically challenging study providing important insight into the importance of the migratory CD11b+ DC subsets in the intestine in context of Th2 responses elicited by gut residing helminths.

The study clearly shows a role for the respective subsets in the small intestine and colon, a mechanistic link between IRF4-deficiency, almost complete ablation of the CD11b+ subsets and reduced antigen transport is missing. Previously, it has been shown that IRF4fl/flCD11cCre mice lack CD11b+CD103+ DCs and IRF4 in DC directly targets the MHC class II presentation machinery (Vander Lugt, Nature Immunol 2014). With the MHC2-dependency of the model in the present study, is it unsurprising that T cell responses are suboptimal when IRF4 is deleted in DCs?

A number of questions arise that must be addressed:

1. Why are bone marrow chimeric mice used? There was no explanation given for such a rationale. As chimeras suffer from intestinal damage as part of the irradiation procedure, while reversible in time - time from creation to use is not clear -with respect to systemic fitness the intestinal alterations are sustained, it is unclear why the authors do not use IRF4fl/flCD11cCre mice.
2. The authors state that 1,000 of the eggs as Th2 antigens are sufficient (p4, line 14). There is no titration of different antigenic doses, an issue maybe as low doses favors Th2 polarization.
3. IRF4fl/flCD11cCre mice delete IRF4 in macrophages (p4, line 38). What is the role of macrophages in this model?
4. What is the viability of transferred DCs and B cells? Can these cells be recovered after the transfer?
5. Cre+ mice have decreased numbers of DC (p7, line 24). Is this related to the expression of the Cre within CD11c+ cells? If so, a Cre- mouse is unsuitable as a control and the authors should provide at least one experiment using a Cre+IRF4fl/wt control.
6. Cre+ mice selectively ablate the CD11b+ DC population, while CD103 SP DCs are not disturbed (Fig. 4e,f). What is relevance as this subset has been shown to suppress Th2 responses (Everts, JEM, 2016) by constitutive expression of IL-12.

Minor:

B cells also take up antigen (p5, line 28). Please provide a plot comparing them to DC to support your statement.

Large parts of the discussion are repetitions of results and conclusions already drawn in the results section. Especially the first 2 paragraphs.

Fig.1:

- What is the impact on Th17 cytokines (Th1/Th17, Th2/Th17)?
- 10x increase in cytokines (compare a and c)
- more variations of cytokine levels between the different panels a,c,d
- weak response against T muris, IL-4 in Cre+ seems to be below detection limit of the ELISA kit

Fig. 2:

- a. is the SEA processed – will the fluorescent dye be degraded during this process?
- b. include macrophages

Fig. 4:

- a. almost 3x increase of CD103 SP DCs (facs plot) but numbers appear the same, while the decrease in DP is reflected in the numbers
- b. since total isolated numbers are the same and DCs are decreased across the board – which cells compensate for the loss?
- d. IL-4 and IL-5 appears significantly decreased – could the authors provide the values?
- d. comparison to C57BL/6 – why not cre- littermates as before?

Fig. 5:

- a. please provide values – does not appear significant
- justification for switching between different statistical tests?

Fig. S2:

- e. no controls shown
- g. cre- and cre+ mixed up

Reviewer #2 (dendritic cells, asthma) (Remarks to the Author):

This is a great story about the heterogeneity that exists in IRF4-dependent cDC2s of the gut. It was already known that IRF4-dependent DCs are needed for Th2 responses in other organs such as the lung, but the authors have done a great job technically to address this in the small intestine and colon. It was already known that IRF4 mainly affects the migratory capacity of cDC2s. However by injecting cDC subsets under the MLN capsule, this is no longer an issue. The authors have also elegantly employed MLN resection to collect cDC subsets from the thoracic duct. It is very seldom that this technique (originally developed in rats and sheep) is employed in mice, but the results are very informative indeed. Finally, also the technique of subserosal injection of eggs is innovative and elegant.

Minor comments

in many organs like the gut and lung, cDC2 subsets can be differentially dependent on IRF4. The expression of CD24 has been used to separate these differentially dependent populations. Have the authors addressed the CD24+ cDC2s separately ?

Reviewer #3 (dendritic cells, system biology) (Remarks to the Author):

The authors study the distinct subsets of intestinal DCs during parasitic worm infections and propose that “distinct subsets of CD11b+ IRF4+ dendritic cells drive Th2 responses in the small intestine and colon”.

Main comments

- Working with a more specific CRE-mouse would have been better. A *zbtb46*-CRE mouse would have done the trick since intestinal macrophages do not express *zbtb46* and are not efficiently targeted by the *zbtb46*-CRE (Loschko et al. J Exp Med 2016). They do however express *CD11c* and this is not an ideal situation. Since a lot is also done by DC transfer some of these experiments allow to zoom in on the role of DCs and this part of the paper is worth a lot. It would however be better to be much more defensive in the way the macrophages are discussed in this manuscript.

- One should be careful with the following sentence: "IRF4 was almost exclusively expressed by intestinal DCs and not by macrophages (Supplementary Fig. 2d-g), suggesting that intestinal DCs were responsible for the impaired phenotype." Transcription factors can play important roles even when they are lowly expressed. Macrophages express lower levels of *IRF8* and *IRF4* as compared to *CD103+* DCs and *CD11b+* DCs, respectively, but *IRF8* still controls an important set of functional genes in macrophages. It could well be that intestinal macrophages lacking *IRF4* are functionally impaired. This is very plausible since *IRF4* controls the expression of many M2-associated genes. One way to address this would be to make competitive chimeras of WT and *CD11c*-CRE-*IRF4*-fl-fl mice (as were performed in Figure 4). Intrinsic effects on the macrophages can then be evaluated. Checking the expression of known M2 genes in macrophages would be a good start. *Relm-a* expression can for example be checked by flow cytometry. Other M2 genes could be checked by RT-PCR.

To demonstrate the fact that DCs are linked to the Th2 induction the authors also perform elegant DC transfers. This is much more convincing than stating that because macrophages have low *IRF4* expression this will probably not be involved.

Two options:

- Adapt the text and add that "albeit macrophages express a lower *IRF4* expression a role in these cells cannot be excluded. To address whether *IRF4* in DCs was necessary for TH2 induction, we performed DC transfers".

- Maintain that *IRF4* has no role in macrophages but demonstrate this by competitive chimeras.

- The authors here propose a major role of *IRF4* in migration of the *CD11b+* DCs. Agace and colleagues (also an author of the present manuscript) previously proposed a similar expression of *CCR7* and a defect in survival rather than migration (Persson et al. Immunity 2013). Could the authors check *CCR7* expression and perform a survival assay? If there is a survival difference then it is surprising that both DCs induce similar T cells responses.

- The title is an overstatement in my view. The current title is "Distinct subsets of *CD11b+* *IRF4+* dendritic cells drive Th2 responses in the small intestine and colon." This is based on the fact that in the small intestine the DCs driving the TH2 response are the *CD103+CD11b+* DCs while in the colon this is performed by the *CD103-CD11b+* DCs. Both cells are closely related developmentally. One could look at it from a different perspective. *CD11b+* DCs are found in the small intestine and the colon (and in most tissues). In the small intestine they acquire the expression of *CD103* but not in the colon. Similarly, in the spleen they acquire the expression of *CD4*, in the lungs they acquire expression of *CD24*. Gene expression profiling of these cells reveals tissue specific signatures (as shown in Figure 3). Nonetheless it is this same subset that induces the TH2 responses in all tissues, regardless of the genes that are differentially expressed due to their tissue of residence and regardless of expression of *CD103*. As the authors state themselves: this is 'in line with previous findings that demonstrate that *CD11b*-expressing DCs from the spleen, lung and skin induce Th2 responses (REFs 5,26,35-37)." The title is thus a bit misleading when one looks at this from a developmental point of view. In fact, there is no evidence that the small intestine (*CD103+*)*CD11b+* DCs induce the TH2 response in a different way than the colon (*CD103-*)*CD11b+* DCs. It could very well be through the exact same mechanisms (maybe *OX40L* among others?). The fact that only 3 genes are common can be very misleading. Genes that are up-regulated 2,2 times in small intestine DCs but 1,9 times in colon DCs (and that do not make the 2-fold cut-off) will not be in the shared list for example... Maybe the mRNA of the relevant TH2 inducing factors are not expressed in the MLN anymore and should have been measured in the DCs while still in the intestine...

REVIEWERS' COMMENTS:

Reviewer #1 (Remarks to the Author):

My queries have been faithfully addressed.

Reviewer #3 (Remarks to the Author):

I thank the authors for adapting the manuscript. The text is now more balanced and in my view more clear.

Reviewer #1 (Helminth, innate) (Remarks to the Author):

In this ms the authors have employed both an elegant and novel approach and do report insightful observations.

The ms describes a well-performed, technically challenging study providing important insight into the importance of the migratory CD11b+ DC subsets in the intestine in context of Th2 responses elicited by gut residing helminths.

The study clearly shows a role for the respective subsets in the small intestine and colon, a mechanistic link between IRF4-deficiency, almost complete ablation of the CD11b+ subsets and reduced antigen transport is missing. Previously, it has been shown that IRF4^{fl/fl}CD11cCre mice lack CD11b+CD103+ DCs and IRF4 in DC directly targets the MHC class II presentation machinery (Vander Lugt, Nature Immunol 2014). With the MHC2-dependency of the model in the present study, is it unsurprising that T cell responses are suboptimal when IRF4 is deleted in DCs?

A number of questions arise that must be addressed:

1. Why are bone marrow chimeric mice used? There was no explanation given for such a rationale. As chimeras suffer from intestinal that damage as part of the irradiation procedure, while reversible in time - time from creation to use is not clear -with respect to systemic fitness the intestinal alterations are sustained, it is unclear why the authors do not use IRF4^{fl/fl}CD11cCre mice.
(NB: This question was highlighted by the editor)

The most efficient way for us to examine the function of IRF4 in CD11c-expressing hematopoietic cells was to perform these experiments using bone marrow chimeras. This enabled us to perform surgical procedures including cannulations and subcapsular injections, which needed to take place in Glasgow, using mice that were bred and maintained in Sweden. Our experience with performing such experiments (see below) indicated that this approach would generate high-quality data.

We understand the reviewers concern about any adverse effect of the irradiation used to generate these chimeras. Experiments with irradiated animals were performed a minimum of 8 weeks after irradiation. The manuscript has been amended to indicate this (p14, line 12). In our experience, adverse effects in the intestine caused by the irradiation subside after a few weeks. In our hands, a sensitive readout for intestinal inflammation is an increased influx of Ly6C^{hi} monocytes (Bain et al., 2014; Bain and Mowat, 2014). No such increase in monocyte populations was observed in cre- or cre+ chimeras when compared to wild type C57BL/6 animals. These data, reproduced below, have been added to the supplementary figures (Supplementary Fig. 2h).

Supplementary Fig. 2h. Representative FACS plots depicting Ly6C^{hi} monocytes and mature Ly6C⁻ MHCII⁺ macrophages in C57BL/6 mice and IRF4^{fl/fl} CD11c-cre positive (cre+) or littermate IRF4^{fl/fl} cre-negative (cre-) BM chimeras

2. The authors state that 1,000 of the eggs as Th2 antigens are sufficient (p4, line 14). There is no titration of different antigenic doses, an issue maybe as low doses favours Th2 polarization.

It has been reported in the literature that *S. mansoni* egg antigens induce robust IFN- γ and Th2 responses at a broad range of concentrations (Pearce and MacDonald, 2002; Pearce et al., 2004). In line with these findings, we observed that the subserosal injection of 500, 1,000 or 2,500 eggs into the intestinal subserosa induced comparable levels of IFN- γ and Th2 responses from restimulated cells. Thus, the dose of injected *S. mansoni* eggs used here should not be considered as 'low'. These data, which reveal no significant differences in the concentrations of cytokines induced, are reproduced below, and have been added to Supplementary Fig. 1e.

Supplementary Fig. 1e. Cytokine responses of restimulated MLN cells from egg injected mice. Different amounts of *S. mansoni* eggs were injected into the subserosal layer of the small intestine and resulting T cell responses were analysed after 5 days by restimulating MLN cells for an additional 3 days in the presence of SEA (n=4-6 mice per group, in two independent experiments, mean \pm SEM, Kruskal Wallis tests (not significant)).

3. *IRF4fl/fl;CD11cCre* mice delete *IRF4* in macrophages (p4, line 38). What is the role of macrophages in this model? (This question was highlighted by the editor)

As we show in Supplementary Fig. 3c, there are no CD64⁺ macrophages (or Ly6C⁺ cells) in the lymph we collect. On the other hand, all DC subsets migrate in the lymph from the intestinal lamina propria to the draining lymph nodes. These observations are consistent with previous publications from our lab and others (Schulz et al., 2009; Cerovic et al., 2013; 2015). Because macrophages, unlike DCs, do not migrate in lymph to the lymph nodes, we did not assess their ability to prime Th2 responses in the lymph nodes. Furthermore, our DC transfer experiments demonstrate that conventional DCs are sufficient for priming Th2 responses and are the dominant cell type priming Th2 responses in our model. While macrophages play many important roles in the intestine, our data clearly demonstrate that they do not migrate and prime Th2 responses to *S. mansoni* eggs. To clarify these points in the manuscript, we have made the following changes: we have made the statement about whether macrophages or DCs are responsible for initiating Th2 responses less dogmatic (p5, line 8-10); and we have added a sentence referring to Supplementary Fig. 3c (p5, line 11-15), where we show that CD64⁺ macrophages are absent from the thoracic duct lymph of our MLNx animals and therefore cannot be responsible for priming lymph node Th2 responses.

4. What is the viability of transferred DCs and B cells? Can these cells be recovered after the transfer?

Both cell viability and purity were >95% after cell sorting for all experiments and the example plots below have been added to the supplementary figures (Supplementary Fig. 3d). While establishing the technique for subcapsular injections for a previous publication from the lab (Cerovic et al., 2015), experiments by flow cytometry and immunohistochemistry were performed to demonstrate that the injected cells could be identified and recovered after these injections. In developing this technique, we were inspired by published experiments where DCs were delivered into lymph nodes and their subsequent migration elegantly tracked (Braun et al., 2011). We have amended the manuscript to provide more detail about the technique (p6, line 3-5).

Supplementary Fig. 3d. Representative sort purity of lymph migrating MHCII^{hi} B220⁻ CD11c^{hi} DCs from MLNx mice.

5. *Cre+ mice have decreased numbers of DC (p7, line 24). Is this related to the expression of the Cre within CD11c+ cells? If so, a Cre- mouse is unsuitable as a control and the authors should provide at least one experiment using a Cre+IRF4fl/wt control.*

We have previously demonstrated that mice with a single allele deletion of IRF4 (IRF4fl/-) show an equivalent phenotype to mice with both alleles of IRF4 present, such as C57BL/6, or Cre- IRF4fl/fl mice (Persson et al., 2013) and similar observations have also been reported by others (Vander Lugt et al., 2014; Williams et al., 2013).

In addition, this strain of CD11c-cre mice has been used extensively for analysis of DC populations and cre expression in these animals does not cause a decrease in the numbers of DCs (Caton et al., 2007; Varol et al., 2009). In our study, we therefore used Cre- IRF4fl/fl as controls for our experiments with Cre+ IRF4fl/fl animals. We have amended the manuscript and included a comment about the fact that CD11c-Cre expression does not affect DC populations (p4 line 42).

6. *Cre+ mice selectively ablate the CD11b+ DC population, while CD103 SP DCs are not disturbed (Fig. 4e,f). What is relevance as this subset has been shown to suppress Th2 responses (Everts, JEM, 2016) by constitutive expression of IL-12.*

In our study we show that CD11b-expressing DCs are sufficient to prime Th2 responses, and that these responses are not initiated in our models where CD11b-expressing DCs are depleted. Everts *et al.* have demonstrated that the CD103 SP DCs are important in regulating and controlling Th2 responses, through their production of IL-12, rather than for priming Th2 responses *per se*. This division of labour between DC populations is an important concept that we have now given more emphasis in the manuscript. Specifically, we have amended the discussion (p10, line 28-31) to clarify this point.

Minor:

B cells also take up antigen (p5, line 28). Please provide a plot comparing them to DC to support your statement.

Our statement refers to data shown in Fig. 2a, that was not clearly described. We apologise, and have corrected the relevant figure reference in the manuscript (p5, line 42).

Large parts of the discussion are repetitions of results and conclusions already drawn in the results section. Especially the first 2 paragraphs.

We have edited the first two paragraphs so they are less repetitive.

Fig. 1:

- What is the impact on Th17 cytokines (Th1/Th17, Th2/Th17)?

We observed no antigen specific Th17 responses induced by *S. mansoni* eggs. This is consistent with published reports of *S. mansoni* and Th17 responses in C57BL/6 mice (Larkin et al., 2012). We have added a comment to this effect (p4, line 18-20)

- 10x increase in cytokines (compare a and c)

Fig. 1a describes the immune responses that were measured after the entire chain of MLNs was harvested. These responses increase when only the relevant individual draining lymph node was harvested (Fig. 1c), a phenomenon that has now been described in more detail in the manuscript (p4, line 27-30)

- more variations of cytokine levels between the different panels a,c,d

Despite some differences in the relative levels of IL-4, IL-5 and IL-13 in panels a,c,d, which may relate to differences in experimental conditions (e.g. the source of cells (as described above) and/or site of injection etc.), the differences between samples and controls are highly significant in all cases.

- weak response against T muris, IL-4 in Cre+ seems to be below detection limit of the ELISA kit

Following this comment, we re-examined the cytokine data from the *T. muris* experiments. The cytokine concentrations in the original version of the manuscript were measured by cytometric bead assay, which is more sensitive than ELISA; this has now been clarified in the Materials and Methods (p15, line 14-21). During our re-examination of these data we realised, however, that some of the values we had reported were below the limit of

detection for the CBA assay, and that we are therefore unable to perform satisfactory statistical analysis to assess the differences in the Th2 response between the cre+ and cre- mice. In the original experiments, we had not only measured IL-4 and IL-13 in restimulation cultures set up d21 post-infection, as we originally reported, but also measured IL-4, IL-9 and IL-13 at d35, by CBA. At this time point, levels of IFN- γ were detectable for 8 out of 9 cre+ and 8 out of 9 cre- samples; levels of IL-4 were detectable for 3 out of 9 cre+ and 0 out of 9 cre- samples; levels of IL-9 were detectable for 9 out of 9 cre+ and 2 out of 9 cre- samples and levels of IL-13 were detectable for 8 out of 9 cre+ and 6 out of 9 cre- samples. Fig. 1f now shows the d35 data for IFN-gamma, IL-9 and IL-13, and the manuscript (p5 line 25-28) has now been modified to describe these data and the numbers of samples in which detectable IL-4 was observed. We hope the reviewer will agree that, despite the fact that we can only detect low levels of IL-4 in this system, our cre+ mice are less able to generate Th2 responses.

Fig. 1f. Secreted cytokines after *T. muris* E/S antigen-specific restimulation of MLN cells 35 days after *T. muris* infection (n=9 mice per group, in three independent experiments, mean \pm SEM, Mann-Whitney U tests, ***P \leq 0.001).

Fig. 2:

- a. is the SEA processed – will the fluorescent dye be degraded during this process?

It is possible that a proportion of the fluorescent dye conjugated to SEA may be degraded during our experiments. The proportions of cells carrying antigen may therefore be higher than we detect. If this degradation is occurring, it is not at a level that prevents us from observing that cre- animals have a higher proportion of AF660⁺ migratory DCs than cre+ mice.

- b. include macrophages

As we have described above, our data and previously published work shows that macrophages do not migrate in murine intestinal lymph. We were therefore not able to assess the transport of antigen by macrophages to the draining lymph nodes

Fig. 4:

- a. almost 3x increase of CD103 SP DCs (facs plot) but numbers appear the same, while the decrease in DP is reflected in the numbers

The reviewer is correct. We show both the numbers and percentages of the DC populations to emphasise this point. Although the percentage of C103 SP DCs increases in the cre+ mice, the absolute numbers of these DCs do not change, as the reciprocal increase in the percentage is due to the decrease of the DP DC population. A similar phenomenon can be observed in Fig. 4e.

- b. since total isolated numbers are the same and DCs are decreased across the board – which cells compensate for the loss?

There are only tens of thousands of DCs in the intestine (Fig. 4b), among 10-20 million total cells (Supplementary Fig. 2c). The decrease of CD11b-expressing DCs in the cre+ mice therefore has no appreciable effect on the total number of intestinal cells. A comment to this effect has been added to p5, line 4-5.

- d. IL-4 and IL-5 appears significantly decreased – could the authors provide the values?

These differences are not significant. Details of the statistics have now been added to the figure legend to clarify this point.

- d. comparison to C57BL/6 – why not cre- littermates as before?

Unfortunately, due to equipment failure only 2 data points could be collected for the cre- group, which would not have allowed any statistical comparison (data not shown). However, cytokine levels fall within the same range as the C57BL/6 data set.

Fig. 5:

- a. please provide values – does not appear significant
- justification for switching between different statistical tests?

We apologize for the confusion caused by using another statistical test. This has been corrected so that tests are used consistently and P-values have been added to the figure legend.

Fig. S2:

- e. no controls shown

IRF4⁺ cells were identified using an isotype control (data not shown). This has now been clarified in the figure legend.

- g. cre- and cre+ mixed up

The figure has been corrected. Please accept our apologies.

Reviewer #2 (dendritic cells, asthma) (Remarks to the Author):

This is a great story about the heterogeneity that exists in IRF4-dependent cDC2s of the gut. It was already known that IRF4-dependent DCs are needed for Th2 responses in other organs such as the lung, but the authors have done a great job technically to address this in the small intestine and colon. It was already known that IRF4 mainly affects the migratory capacity of cDC2s. However by injecting cDC subsets under the MLN capsule, this is no longer an issue.

The authors have also elegantly employed MLN resection to collect cDC subsets from the thoracic duct. It is very seldom that this technique (originally developed in rats and sheep) is employed in mice, but the results are very informative indeed.

Finally, also the technique of subserosal injection of eggs is innovative and elegant.

Minor comments

in many organs like the gut and lung, cDC2 subsets can be differentially dependent on IRF4. The expression of CD24 has been used to separate these differentially dependent populations. Have the authors addressed the CD24+ cDC2s separately?

We and others have observed that CD24 is expressed by the majority of intestinal DCs (Schlitzer et al., 2013; Greter et al., 2012) and cannot be used to distinguish between intestinal DC populations. We show below a typical example of FACS staining for CD24 on lymph migrating DCs, gated as described in the manuscript, to illustrate this point. Here both cDC1s (CD103 SP) and most cDC2s (CD11b SP and DP) are CD24-positive.

Reviewer #3 (dendritic cells, system biology) (Remarks to the Author):

(The editor highlighted the 'general comments' from this reviewer)

The authors study the distinct subsets of intestinal DCs during parasitic worm infections and propose that "distinct subsets of CD11b+ IRF4+ dendritic cells drive Th2 responses in 2 the small intestine and colon".

Main comments

- Working with a more specific CRE-mouse would have been better. A zbtb46-CRE mouse would have done the trick since intestinal macrophages do not express zbtb46 and are not efficiently targeted by the zbtb46-CRE (Loschko et al. J Exp Med 2016). They do however express CD11c and this is not an ideal situation. Since a lot is also done by DC transfer some of these experiments allow to zoom in on the role of DCs and this part of the paper is worth a lot. It would however be better to be much more defensive in the way the macrophages are discussed in this manuscript.

We, and reviewer 1, share this reviewer's concern that it is critically important to carefully distinguish between DCs and macrophages when examining the functions of CD11c+ cells in the intestine. In our response to reviewer 1, we described how we know that macrophages do not migrate in the lymph that drains from the murine intestine, and how we have modified the manuscript to make this clear. We thank this reviewer for also helping us to express this important concept more clearly.

- One should be careful with the following sentence: "IRF4 was almost exclusively expressed by intestinal DCs and not by macrophages (Supplementary Fig. 2d-g), suggesting that intestinal DCs were responsible for the impaired phenotype." Transcription factors can play important roles even when they are lowly expressed. Macrophages express lower levels of IRF8 and IRF4 as compared to CD103+ DCs and CD11b+ DCs, respectively, but IRF8 still controls an important set of functional genes in macrophages. It could well be that intestinal macrophages lacking IRF4 are functionally impaired. This is very plausible since IRF4 controls the expression of many M2-associated genes. One way to address this would be to make competitive chimeras of WT and CD11c-CRE-IRF4-fl-fl mice (as were performed in Figure 4). Intrinsic effects on the macrophages can then be evaluated. Checking the expression of known M2 genes in macrophages would be a good start. Relm-a expression can for example be checked by flow cytometry. Other M2 genes could be checked by RT-PCR. To demonstrate the fact that DCs are linked to the Th2 induction the authors also perform elegant DC transfers. This is much more convincing than stating that because macrophages have low IRF4 expression this will probably not be involved.

Two options:

- Adapt the text and add that "albeit macrophages express a lower IRF4 expression a role in these cells cannot be excluded. To address whether IRF4 in DCs was necessary for TH2 induction, we performed DC transfers".*
- Maintain that IRF4 has no role in macrophages but demonstrate this by competitive chimeras.*

The reviewer has kindly suggested a way for us to resolve the important issue that was raised. In response to this suggestion, which resonates with comments made by reviewer 1, we have not only softened our conclusions based on the IRF4 expression data (p5, line 8-10), but have also added the suggested sentence to the results section of the manuscript. "Although macrophages express a lower IRF4 expression a role in these cells cannot be excluded. ... To further address whether IRF4 in DCs was necessary for TH2 induction, we performed DC transfer experiments." (p5, lines 11-17).

- The authors here propose a major role of IRF4 in migration of the CD11b+ DCs. Agace and colleagues (also an author of the present manuscript) previously proposed a similar expression of CCR7 and a defect in survival rather than migration (Persson et al. Immunity 2013). Could the authors check CCR7 expression and perform a survival assay? If there is a survival difference then it is surprising that both DCs induce similar T cell responses.

We apologise that we have caused confusion here. We do not mean to imply that we have identified a migration defect in our IRF4-deficient DCs. Our data are consistent with data published by Persson *et al.*, in that we see some loss of DP DCs from the intestine, and a larger loss of CD11b SP and DP DCs from the MLN. These data were previously described as indicating a developmental effect of IRF4 loss in the LP, and a survival effect on CD11b-expressing DCs in the MLN. Persson *et al.* also demonstrated that the expression of CCR7 was unaffected in CD11c-cre.IRF4^{fl/fl} (cre+) mice. The additional contribution we make here is to show that despite the effects of IRF4 on the frequencies of DCs, on a per-cell basis, when pulsed with antigen, the remaining DCs are not compromised in their ability to prime antigen-specific CD4 T cell responses. We have amended the manuscript to clarify this point (page 8, lines 6-10 and page 12, line 6-7). We also assessed migrating lymph DCs in cre+ and cre- BM chimeras. However due to the variable yield of cells obtained from each individual cannulated mouse, we were not able to identify whether there are any effects of IRF4-deficiency on the absolute numbers of migrating DC populations in lymph. We did not, however, observe any significant decrease in the

proportion of any of the DC populations compared to their proportions in the LP (data not shown). Thus our data are consistent with IRF4-dependent defects in DC survival in the MLN, rather than on DC migration, being the likely cause for the observed lack of CD11b-expressing DCs in the MLNs. We have amended the manuscript to clarify this point (page 11, lines 38-44).

- The title is an overstatement in my view. The current title is "Distinct subsets of CD11b+ IRF4+ dendritic cells drive Th2 responses in the small intestine and colon." This is based on the fact that in the small intestine the DCs driving the TH2 response are the CD103+CD11b+ DCs while in the colon this is performed by the CD103-CD11b+ DCs. Both cells are closely related developmentally. One could look at it from a different perspective. CD11b+ DCs are found in the small intestine and the colon (and in most tissues). In the small intestine they acquire the expression of CD103 but not in the colon. Similarly, in the spleen they acquire the expression of CD4, in the lungs they acquire expression of CD24. Gene expression profiling of these cells reveals tissue specific signatures (as shown in Figure 3). Nonetheless it is this same subset that induces the TH2 responses in all tissues, regardless of the genes that are differentially expressed due to their tissue of residence and regardless of expression of CD103. As the authors state themselves: this is "in line with previous findings that demonstrate that CD11b-expressing DCs from the spleen, lung and skin induce Th2 responses (REFs 5,26,35-37)." The title is thus a bit misleading when one looks at this from a developmental point of view. In fact, there is no evidence that the small intestine (CD103+)CD11b+ DCs induce the TH2 response in a different way than the colon (CD103-)CD11b+ DCs. It could very well be through the exact same mechanisms (maybe OX40L among others?). The fact that only 3 genes are common can be very misleading. Genes that are up-regulated 2,2 times in small intestine DCs but 1,9 times in colon DCs (and that do not make the 2-fold cut-off) will not be in the shared list for example... Maybe the mRNA of the relevant TH2 inducing factors are not expressed in the MLN anymore and should have been measured in the DCs while still in the intestine...

As the reviewer indicates, we have not been sufficiently careful with our use of the words "subset" and "population". It is not our intention to suggest that the CD11b SP and DP DC populations have different developmental origins. We should not, therefore, have described them as different "subsets". Nevertheless, it is still surprising that, as we show, from the small intestine only DP DCs are sufficient to induce Th2 responses, while in the colon only CD11b SP DCs can perform this function. These different DC populations therefore have different functions in priming Th2 T cells. We have amended the title in response to this comment, and replaced all instances of "subsets" with the more accurate "populations" throughout.

Regarding the results from the microarray experiments, the DCs we used were collected from the lymph, and the SEA-pulsed DCs were demonstrated to be functionally capable of priming SEA-specific Th2 responses *in vivo* (Fig. 3a). We are confident, therefore, that these were a suitable source of material for mRNA analysis. We share the reviewer's concern that small fold-changes in mRNA levels may have important functional effects, but have failed to identify any such changes that might explain at a mechanistic level how these Th2 responses are induced. We have deposited these data in a publically-available repository (Gene Expression Omnibus, accession number GSE91381), and would be very excited if other investigators were able to use them to gain a better understanding of the DC biology that underlies the initiation of Th2 responses.

Additional references

Bain, C.C., A. Bravo-Blas, C.L. Scott, E. Gomez Perdiguero, F. Geissmann, S. Henri, B. Malissen, L.C. Osborne, D. Artis, and A.M. Mowat. 2014. Constant replenishment from circulating monocytes maintains the macrophage pool in the intestine of adult mice. *Nat Immunol.* 15:929–937.

Bain, C.C., and A.M. Mowat. 2014. Macrophages in intestinal homeostasis and inflammation. *Immunol Rev.* 260:102–117.

Braun, A., T. Worbs, G.L. Moschovakis, S. Halle, K. Hoffmann, J. Bölter, A. Münk, and R. Förster. 2011. Afferent lymph-derived T cells and DCs use different chemokine receptor CCR7-dependent routes for entry into the lymph node and intranodal migration. *Nat Immunol.* 12:879–887.

Caton, M.L., M.R. Smith-Raska, and B. Reizis. 2007. Notch-RBP-J signaling controls the homeostasis of CD8-dendritic cells in the spleen. *J Exp Med.* 204:1653–1664.

Cerovic, V., S.A. Houston, C.L. Scott, A. Aumeunier, U. Yrlid, A.M. Mowat, and S.W.F. Milling. 2013. Intestinal CD103(-) dendritic cells migrate in lymph and prime effector T cells. *Mucosal Immunol.* 6:104–113.

Cerovic, V., S.A. Houston, J. Westlund, L. Utriainen, E.S. Davison, C.L. Scott, C.C. Bain, T. Joeris, W.W. Agace,

- R.A. Kroczek, A.M. Mowat, U. Yrlid, and S.W.F. Milling. 2015. Lymph-borne CD8 α + dendritic cells are uniquely able to cross-prime CD8+ T cells with antigen acquired from intestinal epithelial cells. *Mucosal Immunol.* 8:38–48.
- Greter, M., J. Helft, A. Chow, D. Hashimoto, A. Mortha, J. Agudo-Cantero, M. Bogunovic, E.L. Gautier, J. Miller, M. Leboeuf, G. Lu, C. Aloman, B.D. Brown, J.W. Pollard, H. Xiong, G.J. Randolph, J.E. Chipuk, P.S. Frenette, and M. Merad. 2012. GM-CSF controls nonlymphoid tissue dendritic cell homeostasis but is dispensable for the differentiation of inflammatory dendritic cells. *Immunity.* 36:1031–1046.
- Larkin, B.M., P.M. Smith, H.E. Ponichtera, M.G. Shainheit, L.I. Rutitzky, and M.J. Stadecker. 2012. Induction and regulation of pathogenic Th17 cell responses in schistosomiasis. *Semin Immunopathol.* 34:873–888.
- Pearce, E.J., and A.S. MacDonald. 2002. The immunobiology of schistosomiasis. *Nat Rev Immunol.* 2:499–511.
- Pearce, E.J., C. M Kane, J. Sun, J. J Taylor, A.S. McKee, and L. Cervi. 2004. Th2 response polarization during infection with the helminth parasite *Schistosoma mansoni*. *Immunol Rev.* 201:117–126.
- Persson, E.K., C.L. Scott, A.M. Mowat, and W.W. Agace. 2013. Dendritic cell subsets in the intestinal lamina propria: ontogeny and function. *Eur J Immunol.* 43:3098–3107.
- Schlitzer, A., N. McGovern, P. Teo, T. Zelante, K. Atarashi, D. Low, A.W.S. Ho, P. See, A. Shin, P.S. Wasan, G. Hoeffel, B. Malleret, A. Heiseke, S. Chew, L. Jardine, H.A. Purvis, C.M.U. Hilken, J. Tam, M. Poidinger, E.R. Stanley, A.B. Krug, L. Renia, B. Sivasankar, L.G. Ng, M. Collin, P. Ricciardi-Castagnoli, K. Honda, M. Haniffa, and F. Ginhoux. 2013. IRF4 transcription factor-dependent CD11b+ dendritic cells in human and mouse control mucosal IL-17 cytokine responses. *Immunity.* 38:970–983.
- Schulz, O., E. Jaensson, E.K. Persson, X. Liu, T. Worbs, W.W. Agace, and O. Pabst. 2009. Intestinal CD103+, but not CX3CR1+, antigen sampling cells migrate in lymph and serve classical dendritic cell functions. *J Exp Med.* 206:3101–3114.
- Vander Lugt, B., A.A. Khan, J.A. Hackney, S. Agrawal, J. Lesch, M. Zhou, W.P. Lee, S. Park, M. Xu, J. DeVoss, C.J. Spooner, C. Chalouni, L. Delamarre, I. Mellman, and H. Singh. 2014. Transcriptional programming of dendritic cells for enhanced MHC class II antigen presentation. *Nat Immunol.* 15:161–167.
- Varol, C., A. Vallon-Eberhard, E. Elinav, T. Aychek, Y. Shapira, H. Luche, H.J. Fehling, W.-D. Hardt, G. Shakhar, and S. Jung. 2009. Intestinal lamina propria dendritic cell subsets have different origin and functions. *Immunity.* 31:502–512.
- Williams, J.W., M.Y. Tjota, B.S. Clay, B. Vander Lugt, H.S. Bandukwala, C.L. Hrusch, D.C. Decker, K.M. Blaine, B.R. Fixsen, H. Singh, R. Sciammas, and A.I. Sperling. 2013. Transcription factor IRF4 drives dendritic cells to promote Th2 differentiation. *Nat Commun.* 4:2990.